# Boosting Document Layout Analysis with Graphic Multi-modal Data Fusion and Spatial Geometric Transformation

## Abstract

Document layout analysis is essential for Document Intelligence, playing a pivotal role in automated understanding and processing of document content. Most existing approaches within this domain are predicated on computer vision techniques that concentrate on image modality, despite documents containing both rich visual and textual information. While recent advances in multi-modal approaches begin to incorporate word embeddings to enhance recognition capabilities, they also incur a substantial computational burden. Moreover, the diversity of document structures demands models with great robustness, especially during the document editing process. In this paper, we introduce pluggable and efficient data pre-processing strategies to boost the layout analysis performance. Firstly, we discover that element categories depend on relative relationships and propose a Graphical Multi-modal Data Fusion technique, which constructs a graph to establish connections between disparate textual segments. Secondly, in terms of structural diversity of documents, we devise a Spatial Geometric Transformation strategy to improve model robustness against layout alterations. Our methods operate during the pre-processing phase, which facilitates straightforward integration with existing models to achieve significant accuracy increase with negligible extra computations. Experimental results show that our strategies illustrate State-Of-The-Art performance across multiple document layout analysis datasets.

## 1 Introduction

Document Intelligence encompasses the ability to understand, analyze, and process document content with deep learning methods(Cui et al., 2021). It includes sub-tasks such as Document Layout Analysis (DLA)(Pfitzmann et al., 2022; Cheng et al., 2023; Shi et al., 2023; Ma et al., 2023; Chen et al., 2024), Document Classify(Harley et al., 2015; Larson et al., 2022; Naeem et al., 2022), Key Information Extraction (KIE)(Jaume et al., 2019; Xu et al., 2022; Luo et al., 2023), Table Recognition(Li et al., 2020b; Abdallah et al., 2022; Long et al., 2024), Optical Character Recognition (OCR)(Nayef et al., 2017; Shi et al., 2016; Li et al., 2023) , Visual Question Answering (VQA)(Mathew et al., 2021; Tanaka et al., 2021; Ishmam et al., 2024; Hu et al., 2024), etc. Among them, DLA stands as a fundamental branch, tasked with detection and recognition of various textual, graphical, and tabular elements from documents. There are multiple application scenarios of DLA, including information retrieval, archive digitization, etc. Besides, one of the main ways to acquire training data for Large Language Models (LLMs) is to obtain structured corpus from books, papers, newspapers, etc., which are stored in PDF or Word documents(Liu et al., 2024; Weber et al., 2023). Therefore, it is crucial to analyze the layout of original documents and extract texts to increase the quality of training data for LLMs. Nonetheless, DLA is a challenging task in terms of the diversity of document structures. The occurrence of a minor error significantly disrupts the integrity of the overall extracted architecture, leading to misinterpretations of hierarchical structure and flow of information.

Modern DLA models can be divided into two approaches: Vision-based and Multi-modal. The first is putting DLA as a computer vision task, where the primary frameworks are object detectors(Cheng et al., 2023) and instance segmentation models(Lee et al., 2019). The other approach utilizes both visual and textual features(Zhang et al., 2021; Da et al., 2023). Different from computer vision tasks, DLA benefits greatly from the textual semantic information and the integration of both modalities

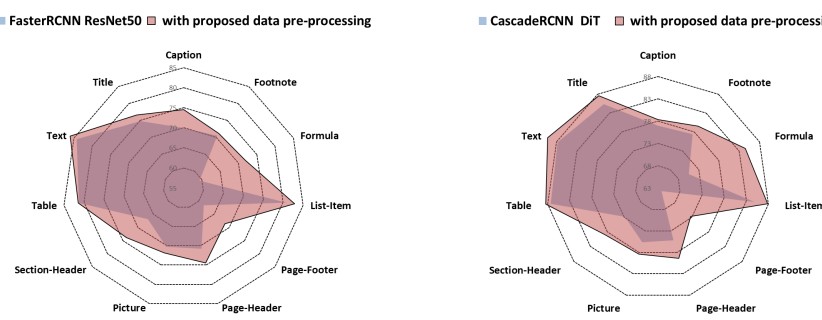

Figure 1: Comparsion of mAP by category on DocLayNet.

promotes the ability to detect and recognize document components (Yu et al.). Compared with Vision-based methods, Multi-modal DLA models possess higher accuracy, but require more computation, as the complexity of handling multiple modal inputs for feature extraction.

As for DLA, we observe that the relative relationships between text elements play a decisive role. The category of a sentence is influenced not just by its intrinsic features, but also by the semantics, position, and visual characteristics of associated texts. For instance, relocating a "Page-header" to the bottom of a document will let DLA model reclassify it as a "Page-footer". Therefore, establishing connections between sentences is beneficial, as it enables models to more accurately comprehend document structures. Graph neural networks(Kipf & Welling, 2016; Velickovic et al., 2017) are effective in explicitly modeling node relationships(Han et al., 2022b), so it is promising to construct graphic connections in DLA task. In addition, documents often experience random positional alterations of sentences during editing and it is crucial to integrate geometric transformation techniques to simulate the above changes, thereby increasing generalization to layout disturbances.

In this paper, we put forward efficient and pluggable data pre-processing approaches for DLA to enhance recognition performance. On one hand, we propose a Graphic Multi-modal Data Fusion method that fosters sentence connectivity. We firstly employ parsing plugins, such as pdfplumber[1], python-docx[2], OCR, etc., to extract sentences and words from original documents, and utilize a tokenizer for words tokenization and acquiring word embeddings. Inspired by Transformer-iN-Transformer(Han et al., 2021; 2022a), we construct a word graph for each sentence to integrate fine-grained semantic features and set sentences as nodes to construct a sentence graph to build relationships. Finally, we fuse the words, sentences, and image information for object detection. On the other hand, we propose the Spatial Geometric Transformation at sentence, paragraph and page levels to facilitate DLA models to handle the subtle differences of document layout. To increase semantic diversity in sentence dimension, we propose a remix operation to promote sentence combinations. For paragraph level, we put up scale and translate operations to change document structures. For the entire page level, we randomly crop them to change the global characteristics.

Our proposed data pre-processing methods can be directly combined with various types of object detectors with negligible increases in computational cost. As Figure 1 shows, we improve the accuracy of all categories compared with baseline performance, Even for elements that rely more on image features such as *Picture* and *Table*. Compared with existing SOTA models VGT(Da et al., 2023) on DocLayNet(Pfitzmann et al., 2022), we achieve higher accuracy with almost half computations ("**82.4** mAP, **643.7G** FLOPS" vs. "**81.2** mAP, **1248.9G** FLOPS").

## 2 RELATED WORK

We divide DLA models into two types of approaches: Vision-based and Multi-modal. The overall architecture is depicted in Figure 2.

---

[1]https://github.com/jsvine/pdfplumber

[2]https://github.com/python-openxml/python-docx

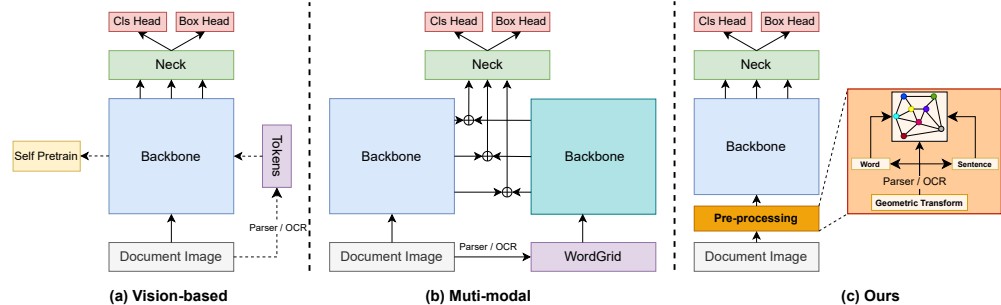

Figure 2: Comparion of different framework of DLA models. The dashed line in the left subplot signifies the use of textual semantic information for pre-training purposes, which is not employed during inference.

**Vision-based models.** These methods put DLA as pure vision tasks, as shown in Figure 2(a). Li et al.(Li et al., 2020a) treated DLA as an object detection task and introduced a domain adaptation module to address cross-domain document detection. Cheng et al.(Cheng et al., 2023) proposed TransDLANet, which utilizes an adaptive element matching mechanism that aligns the query embedding more closely with the ground truth. Different from object detection, Lee et al.(Lee et al., 2019) tackled the DLA task with a segmentation approach and incorporated a trainable multiplicative layer technique to enhance the accuracy of object boundary detection, thereby improving the performance of pixel-level segmentation networks.

With the development of self-supervised pretraining, document intelligence models also shift from "supervised learning" to a paradigm of "self-supervised pretraining to fine-tuning". The LayoutLM series models(Xu et al., 2020b;a; Huang et al., 2022), Unidoc(Gu et al., 2021), Selfdoc(Gu et al., 2024) constructed a document level pre-training method to utilize images and textual semantic features as well as layout relationships for and pre-training. However, when these models are applied to downstream DLA tasks, they only consider visual features without textual information, which leads to significant gaps in pre-training and fine-tuning. Therefore, they are essentially Vision-based methods, as shown by the dashed lines in Figure 2(a).

**Muti-modal models.** The other type of DLA models are of Muti-modal methods, as shown in Figure 2(b). Wordgrid represents that this image is a semantic image composed of word embedding, and its spatial size is consistent with the image. Yang et al.(Yang et al., 2017) treated document semantic structure extraction as a pixel-wise segmentation task. The proposed model classifies pixels based on both visual features and sentence-level grids to elevate the precision of layout analysis. Zhang et al. (Zhang et al., 2021) proposed VSR, which integrates visual, semantic, and relational information to significantly improve detection accuracy. Da et al.(Da et al., 2023) put forward a two-stream model VGT. The two-stream framework includes a Vision Transformer (ViT) for encoding image features and a Grid Transformer (GiT) for encoding textual features based on a 2D token-level grid representation of the document. Besides, this method comprises two pre-training methods called Masked Grid Language Modeling (MGLM) and Segment Language Modeling (SLM). VGT achieves SOTA performance on multiple DLA benchmarks.

Although Multi-modal approaches offer superior accuracy over Vision-based methods, the higher computational cost and parameters cannot be ignored. Current Multi-modal DLA models often involve an additional large-scale network to extract textual features, almost doubling the computational load. To address these challenges, we propose the efficient data pre-processing techniques that significantly enhance the accuracy while minimally increasing computational requirements, as shown in Figure 2(c).

## 3 OUR METHODS

In this section, we introduce the proposed idea of utilizing data pre-processing to improve the effectiveness of layout analysis. The idea includes Multi-modal data fusion based on graph structure and data augmentation based on spatial geometric transformation.

### 3.1 GRAPHIC MULTI-MODAL DATA FUSION

As we preliminarily introduced, the categories of textual elements are contingent on their relative positions. We represent this relation with a graph, where sentences act as nodes and construct edges according to the coordinates. Additionally, recognizing that each sentence is comprised of words, we devise a strategy that combines coarse-grained sentence-level connections with fine-grained word-level relationships.

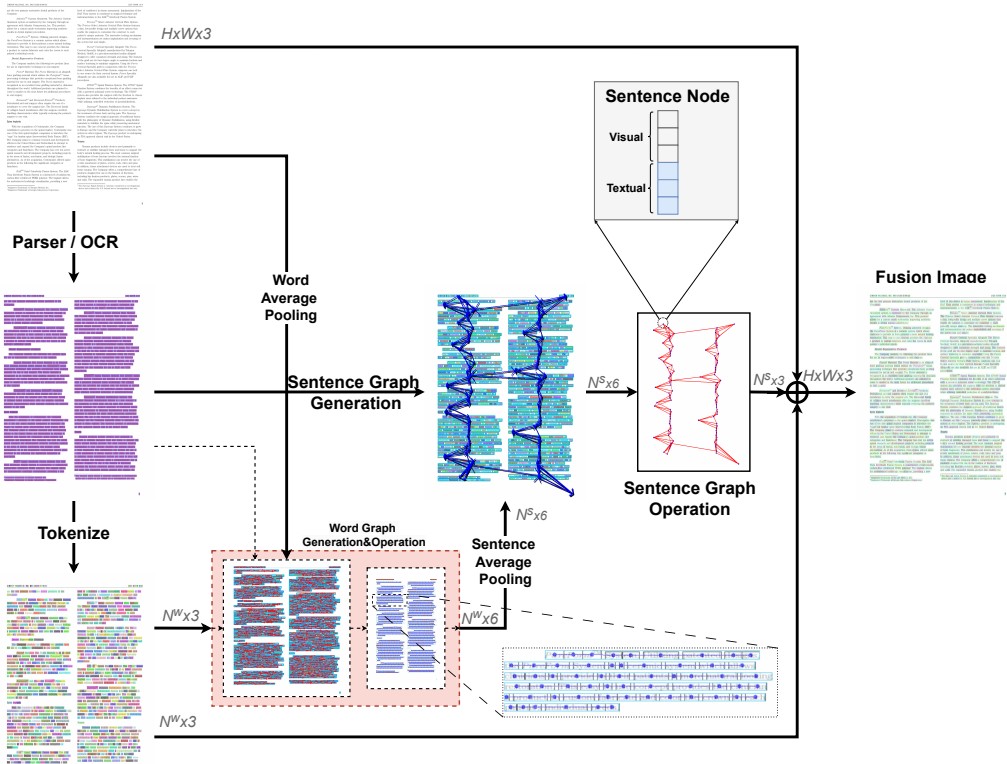

Figure 3: Illustration of Graphic Multi-modal Data Fusion. The red dashed area is an optional operation for constructing a word graph within each sentence. The shape of each input is also displayed in the figure.

Initially, we process a document image $\mathbf{I} \in \mathbb{R}^{H \times W \times 3}$ to directly extract the sentence coordinates, word coordinates and recognize words with parsing plugins or OCR. Subsequently, we utilize the approach proposed by Denk et al.(Denk & Reisswig, 2019), employing the "bert-base-multilingual-uncased" model[3] to tokenize words into sub-word tokens and obtain word embeddings. The width of word boxes is equally split for each token. To streamline the subsequent graph construction process and minimize computational demands, we introduce a fully connected layer to project the original 768-dimensional word embeddings into a reduced 3-dimensional space.

The first part is the construction of word graph within each sentence. Each token is set as a node, with 6-dimensional feature obtained by concatenating 3-dimensional textual and the mean of 3-dimensional token image information. We sort the tokens in each sentence from left to right. For the $m$ token $w_m$ in sentence $i$, it needs to be connected to the $(m-1)$ token $w_{m-1}$ (left) and the $(m+1)$ token $w_{m+1}$ (right). So the construction of word graph $E_i^w$ within sentence $i$ can be expressed as:

$$E_i^w = \{(w_m, w_{m+1}) \mid w_m, w_{m+1} \in \omega_i, m = 1, \ldots, |\omega_i|-1\} \cup \{(w_m, w_{m-1}) \mid w_m \in \omega_i, m = 2, \ldots, |\omega_i|\} \tag{1}$$

---

[3]https://huggingface.co/google-bert/bert-base-multilingual-uncased

where $\omega_i$ is the set of tokens in sentence $i$.

The second is the construction of sentence graph. We treat each sentence as a node whose information is the mean of features of tokens. For each node, we find the 4 nearest sentences to construct edges. We define $S$ as the set of all sentences, and the $i$ sentence can be represented as a point set $P_i$. The calculation of the closest distance between $P_i$ and $P_j$ is as follows:

$$D_{ij} = \min_{(m \in P_i, n \in P_j)} \sqrt{(x_m - x_n)^2 + (y_m - y_n)^2} \tag{2}$$

where $(x_m, y_m)$ and $(x_n, y_n)$ are the coordinates on points $m \in P_i$ and $n \in P_j$. We select the 4 sentences with the smallest distance to form the nearest sentence set $S_{near}^i$.

$$S_{near}^i = \{(P_{i_1}, P_{i_2}, P_{i_3}, P_{i_4}) | (i_1, i_2, i_3, i_4) \in arg\,min\,\{D_{ij} | j \in \{1, 2, \ldots, |S|\}, j \neq i\}\} \tag{3}$$

The edge construction detail code is shown in Appendix A.5.

After establishing the edges of word and sentence, the subsequent section elaborates on the graph attention calculation. Since the calculation method for sentence graph and word graph is same, we take sentence graph calculation as example. The process is formalized as:

$$\mathbf{S}' = \mathbf{G}^S(\mathbf{S}, \mathbf{E}^S) \tag{4}$$

Here, $\mathbf{G}^S$ denotes the graph operation, and $\mathbf{E}^S$ represents the edges connecting the nodes within the sentence graph. The input sentence information $\mathbf{S}$ is a matrix in $\mathbb{R}^{N \times d}$, where $N$ is the total number of sentences and $d$ is the combined feature dimensionality, setting as 6.

For any pair of sentences $i$ and $j$, we calculate the unnormalized attention score $e_{ij}$ utilizing a feed-forward neural network with a single hidden layer, as follows:

$$e_{ij} = \text{LeakyReLU}\left(\mathbf{a}^T\left[\mathbf{W} \cdot \mathbf{s}_i \| \mathbf{W} \cdot \mathbf{s}_j\right]\right) \tag{5}$$

Here, $\mathbf{s}_i$ and $\mathbf{s}_j$ represent the node features of sentences $i$ and $j$, $\mathbf{W}$ is the weight matrix, $\mathbf{a}$ is the weight vector, and $\|$ signifies concatenation. The attention coefficients $\alpha_{ij}$ are normalized by applying a softmax function across all neighboring nodes $N_i$:

$$\alpha_{ij} = \frac{\exp(e_{ij})}{\sum_{k \in N_i} \exp(e_{ik})} \tag{6}$$

Then, we weight the connected node features and project the 6-dimensional representation of each node onto a 3-dimensional space with another fully connected layer, parameterized by $\mathbf{W}_1$:

$$\mathbf{s}_i' = \mathbf{W}_1 \cdot \left(\sum_{j \in N_i} \alpha_{ij}\mathbf{s}_j\right) \tag{7}$$

Finally, we fuse the information of both sentences graphic features and word embedding to the image for Multi-modal data fusion, which can be represented by the following equation:

$$\mathbf{I}'(x,y) = \begin{cases} \mathbf{I}(x,y) + \mathbf{s}_i' + \boldsymbol{w}_m & \text{if } (x,y) \text{ in token } w_m \text{ position of sentence } i \\ \mathbf{I}(x,y) + \mathbf{s}_i' & \text{if } (x,y) \text{ in blank position of sentence } i \\ \mathbf{I}(x,y) & \text{otherwise} \end{cases} \tag{8}$$

where $\boldsymbol{w}_m$ is the 3-dim embedding projected from the original 768-dim token embedding of $w_m$. For any pixel within each sentence, we directly merge the information fused by the sentence graph with the original image information. Considering the existence of gaps between words in each sentence, we additionally introduce the original word embedding to distinguish between word regions (foreground) and gaps (background) in the sentence. This operation can further improve accuracy.

Our proposed module seamlessly integrates into the pre-processing stage. Since the image and textual space dimensions in the graphs are all set as 3, this data fusion module introduces a negligible increase in computational complexity. The experimental results conclusively show that this strategy markedly improves recognition accuracy.

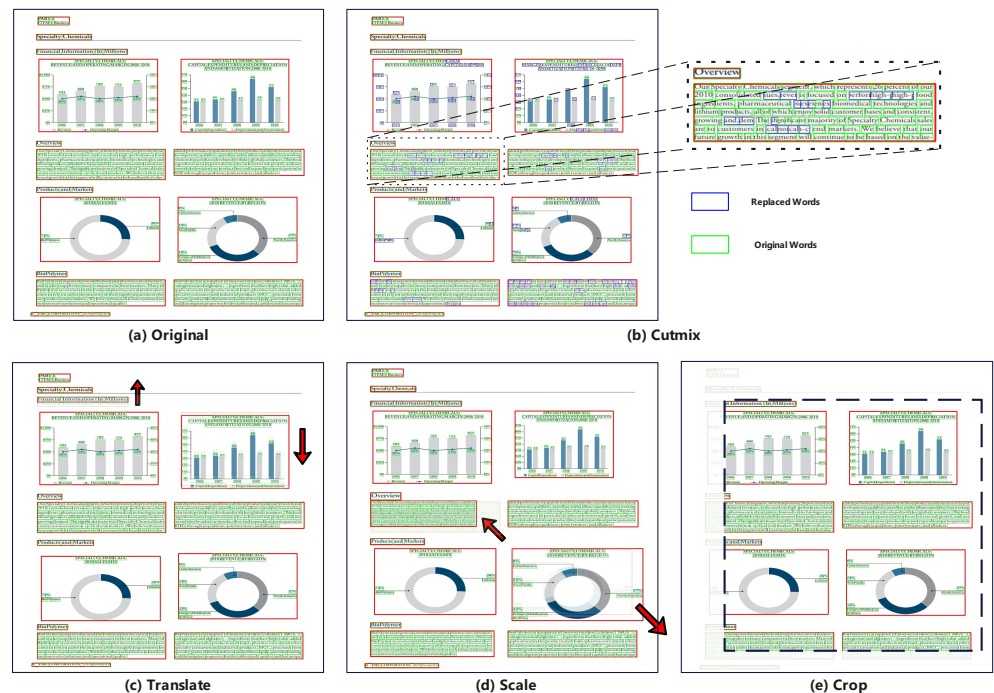

Figure 4: Illustration of Spatial Geometric Transformation.

## 3.2 SPATIAL GEOMETRIC TRANSFORMATION

Regular data enhancement methods usually operate on entire images, which cannot alter the overall arrangement of documents, nor can they change the layout of sentences and paragraphs. For example, no matter how we resize the page image, the relative position between paragraphs remains unchanged. We are also unable to modify the texts to generate a new paragraph. This highlights the limitation of regular approaches, which maintain the fixed layout of textual elements and do not account for dynamic adjustments within the structures. Document layout analysis primarily focuses on learning the structural characteristics of documents, especially the interrelationships between elements. As shown in Figure 4, our main contribution is to construct a strategy to synthesize new image structures of documents, which transitions from sentence-level, through paragraph-level, and ultimately to page-level.

For document spatial augmentation, arbitrary changes may result in illogical pages, thereby generating noisy data that could hurt the precision. To mitigate this, we introduce tailored constraint conditions at each level of data augmentation. We have also redefined the operations involved and the detailed description is as follows.

**Sentence Remix.** For the first, we propose sentence remix to enhance diversity by selectively replacing original word with another one from a distinct sentence. It is crucial to ensure that the newly synthesized document images are plausible. For instance, replacing a larger font title word with a smaller font text word could lead to image distortion during the resizing process. Therefore, any word replacement must meet two criteria to maintain the integrity and realism of the document image: (1) It belongs to the same class as the origin. (2) The box scale difference is within 5%. We use $(x_1^o, y_1^o, x_2^o, y_2^o)$ as the original word position and $(x_1^t, y_1^t, x_2^t, y_2^t)$ as the target word position. For document image $I$, this process can be described as the following:

$$I[y_1^o : y_2^o, x_1^o : x_2^o, :] = Resize(I[y_1^t : y_2^t, x_1^t : x_2^t, :]) \quad where$$
$$max(\frac{max(y_2^t - y_1^t, y_2^o - y_1^o)}{min(y_2^t - y_1^t, y_2^o - y_1^o)}, \frac{max(x_2^t - x_1^t, x_2^o - x_1^o)}{min(x_2^t - x_1^t, x_2^o - x_1^o)}) < 1.05 \quad (9)$$

**Paragraph Perturbation.** We construct paragraph-level variations to simulate the inevitable perturbations during the document editing process, primarily including Scaling and Translation of paragraphs. These perturbations are not casual and must meet the following conditions: (1) The changes should not be too drastic. (2) There should be no Intersection over Union (IoU) between different layout elements. We describe the point position as $p(x, y) = [x, y, 1]^T$ in paragraph $A$. The transformed $\tilde{A}$ is written as follows:

$$\tilde{A} = [\tilde{p}(x_i', y_i') = \tilde{P} \times p(x_i, y_i) \mid \forall(x_i, y_i) \in A]$$

where $\tilde{P}$ is the matrix. We use $\tilde{P}_T$ and $\tilde{P}_S$ to represent Translation and Scaling matrices, as follows:

$$\tilde{P}_T = \begin{bmatrix} 1 & 0 & t_x \\ 0 & 1 & t_y \\ 0 & 0 & 1 \end{bmatrix} \quad \tilde{P}_S = \begin{bmatrix} s_x & 0 & 0 \\ 0 & s_y & 0 \\ 0 & 0 & 1 \end{bmatrix}$$

$t_x$ and $t_y$ denote the pixel value for horizontal and vertical translation, randomly selected within a range of [-2%,2%] of the width and height of $A$. That means $A$ has the potential to translate by a maximum distance of 2% of box width or height in four directions. $s_x$ and $s_y$ are randomly selected within a range of [80%, 120%] of the width and height, which indicates $A$ has the potential to enlarge or shrink by 20% based on the top left corner point.

**Page Clip.** Ultimately, we develop a method for clipping the entire pages. We keep the aspect ratio of the clipped region exactly with the original image $I \in R^{H \times W \times 3}$. The primary purpose is to maintain the visual proportions, making the cropped document images appear more natural and harmonious visually. Additionally, it helps to prevent content distortion, avoiding the stretching or compression of texts due to inconsistent aspect ratios, ensuring that elements retain their original proportions, and readability. The page clip is as follows:

$$\tilde{I} = I[y_s : y_e, x_s : x_e, :] \quad where \quad \frac{y_e - y_s}{x_e - x_s} = \frac{H}{W}$$

where $x_s$, $y_e$, $y_s$, $y_e$ are the crop points of the image.

The proposed Spatial Geometric Transformation is a universal method for layout analysis tasks, which significantly improves detection accuracy on multiple models without adding any parameters or changing the model structure.

## 4 EXPERIMENTS

### 4.1 DATASETS AND IMPLEMENTATION DETAILS

To demonstrate the effectiveness and universality of our method, we conduct experiments on four DLA datasets (DocLayNet(Pfitzmann et al., 2022), D$^4$LA (Da et al., 2023), PubLayNet(Zhong et al., 2019), Docbank(Li et al., 2020c)) with two object detection models (FasterRCNN(Ren et al., 2015) based on a ResNet50(He et al., 2016) backbone, CascadeRCNN(Cai & Vasconcelos, 2018) with DiT(Li et al., 2022) backbone). Please refer to **Appendix A.1** for a detailed introduction of the datasets.

The experimental equipment is $8 \times$ NVIDIA V100 GPUs with 32G graphics memory. Our experimental framework is built upon Detectron2(Wu et al., 2019). For DocLayNet, the training configuration includes a batch size of 32 and a total of 32,000 training steps. For the training of FasterRCNN, we use the SGD optimizer with an initial learning rate of 2e-2, reducing it by a factor of 0.1 at the 20,000th and 26,000th steps. We also follow the common practice by initializing the model with weights pre-trained on the Microsoft COCO 2017 benchmark(Lin et al., 2014). For CascadeRCNN, we opt for the AdamW optimizer(Loshchilov & Hutter, 2017), setting the initial learning rate to 2e-4 following DiT(Li et al., 2022) specifications. Experiments on the D$^4$LA, PubLayNet, DocBank datasets are performed with CascadeRCNN, using a batch size of 32 and an initial learning rate of 2e-4, with training steps totaling 12,000, 120,000 and 200,000, respectively.

For performance assessment, we rely on the mean Average Precision (mAP) at IoU intervals [0.50:0.95:0.05] as our primary metric for accuracy. To evaluate computational efficiency, we also measure Floating Point Operations Per Second (FLOPS) under an 800×800 input image.

## 4.2 ABLATIONS

We conduct ablation experiments on the proposed module based on the two detectors and the DocLayNet dataset.

**Analysis of Graphic Multi-modal Data Fusion.** We first conduct ablation experiments on the Graphic Multi-modal Data Fusion, and the results are shown in Table 1. For FasterRCNN with ResNet50, we utilize the proposed Sentence Graph to establish the connection between dfferent sentences, the mAP is significantly boosted to 74.4. For words inside sentences, we construct a fine-grained graph structure and achieve a 0.6 mAP improvement, which confirms that WordGraph promotes further correlation of internal text semantics. On the contrary, if only WordEmbedding is added to the original image without the proposed graph, its mAP only improves by 0.2 compared to the baseline. This indicates that the proposed graph is the most significant contribution to improving the effectiveness of layout analysis. However, the combination of WordEmbedding can further improve the overall mAP to 75.2, which validates that we use it to distinguish foreground and background in sentences. Similarly, for Cascade R-CNN with DiT, we employ Graphic Multi-modal Data Fusion to significantly improve the mAP of baseline 78.0 to 82.1.

Table 1: Ablation study of Graphic Multi-modal Data Fusion on DocLayNet test set.

| Detector | Backbone | SentenceGraph | WordGraph | WordEmbedding | mAP |
|---|---|---|---|---|---|
| FasterRCNN | ResNet50 | | | | 71.8 |
| | | ✓ | | | 74.4 |
| | | ✓ | ✓ | | 75.0 |
| | | | | ✓ | 72.0 |
| | | ✓ | ✓ | ✓ | **75.2** |
| CascadeRCNN | DiT | | | | 78.0 |
| | | ✓ | | | 80.9 |
| | | ✓ | ✓ | | 81.7 |
| | | | | ✓ | 78.6 |
| | | ✓ | ✓ | ✓ | **82.1** |

**Analysis of projected dimensions.** We project 768 dimensional token embeddings to 3 dimensions to reduce computation. In terms of the information loss in this process, we conduct experiments based on Faster-RCNN and DocLayNet. The results are as shown in Table 2. With the increase of dimension, the mAP has hardly improved, while the FLOPS has increased significantly, even the Flops of 128 dim is 1.5 times that of base model. Therefore, it's to use graphs and attention mechanism to construct the connections between words and sentences, the dim for data fusion does not need to be very large, as the backbone will extract high-level features based on the connections.

Table 2: Ablation study of projected dimensions with Faster-RCNN on DocLayNet test set.

| Method | GMDF-dim | mAP | FLOPS(G) |
|---|---|---|---|
| Faster-RCNN | - | 71.8 | 134.4 |
| Faster-RCNN | 3 | 75.2 | 135.9 |
| Faster-RCNN | 64 | 75.2 | 166.1 |
| Faster-RCNN | 128 | 75.3 | 197.6 |

**Spatial Geometric Transformation Analysis.** We conduct ablation studies to assess the impact of Spatial Geometric Transformation on performance, with results detailed in Table 3. We perform geometric transformations on the original image through Sentence Remix, Pargraph Perturbation and Page Clip, resulting in mAP improvements of 0.4, 0.4 and 0.3, respectively. After integrating these techniques, we observed a cumulative enhancement in mAP to 78.7 from the original baseline. Similarly, FasterRCNN experiences a boost in mAP to 72.6 with Spatial Geometric Transformation.

**Analysis of combination of proposed methods.** We synthesize all our proposed data pre-processing techniques to maximize performance gains, as evidenced in Table 4. For FasterRCNN, the baseline

Table 3: Ablation study of Spatial Geometric Transformation on DocLayNet test set.

| Detector | Backbone | Sentence Remix | Pargraph Perturbation | Page Clip | mAP |
|----------|----------|:--------------:|:---------------------:|:---------:|-----|
| FasterRCNN | ResNet50 | | | | 71.8 |
| | | ✓ | | | 72.2 |
| | | | ✓ | | 72.2 |
| | | | | ✓ | 72.0 |
| | | ✓ | ✓ | ✓ | **72.6** |
| CascadeRCNN | DiT | | | | 78.0 |
| | | ✓ | | | 78.4 |
| | | | ✓ | | 78.4 |
| | | | | ✓ | 78.3 |
| | | ✓ | ✓ | ✓ | **78.7** |

model without our proposed data pre-processing methods achieves 71.8 mAP. The implementation of our Graphical Multi-modal Data Fusion strategy elevates this to 75.2, an improvement that introduces a modest increment of 1.5 GFLOPS, which is negligible in comparison to the original 134.4 GFLOPS. Moreover, the incorporation of Spatial Geometric Transformation, which entails no additional parameters, yields a further 0.8 mAP increase. When these methods are unified, we achieve an accuracy improvement of 4.0 mAP and only increase the computational load by 1.5G FLOPS. Consistently, our experiments with CascadeRCNN reveal a marked improvement in mAP to 82.4, at the cost of 1.5G FLOPS increment. These results not only substantiate the effectiveness of our methods but also highlight their efficiency and pluggability.

Table 4: Ablation study of proposed methods combination on DocLayNet test set.

| Detector | Backbone | SGT[*] | GMDF | mAP | $\Delta_{mAP}$ | FLOPS(G) |
|----------|----------|:------:|:----:|-----|------------|----------|
| FasterRCNN | ResNet50 | | | 71.8 | - | 134.4 |
| | | ✓ | | 72.6 | ↑0.8 | 134.4 |
| | | | ✓ | 75.2 | ↑3.4 | 135.9 |
| | | ✓ | ✓ | **75.8** | ↑**4.0** | 135.9 |
| CascadeRCNN | DiT | | | 78.0 | - | 642.2 |
| | | ✓ | | 78.7 | ↑0.7 | 642.2 |
| | | | ✓ | 82.1 | ↑4.1 | 643.7 |
| | | ✓ | ✓ | **82.4** | ↑**4.4** | 643.7 |

[*] SGT is the abbreviation of "Spatial Geometric Transformation"

### 4.3 COMPARSION WITH OTHER METHODS

We present a comprehensive comparison of our proposed methods with other contemporary approaches on the DocLayNet test set, as shown in Table 5. We divide the other methods into two groups: Vision-based and Multi-modal, reflecting the different strategies employed in DLA. The Vision-based group includes methods like FasterRCNN(Ren et al., 2015), MaskRCNN(He et al., 2017), YoloV5x6(Pfitzmann et al., 2022), CascadeRCNN(Cai & Vasconcelos, 2018), TransD-LANet(Cheng et al., 2023), SwinDocSegmenter(Banerjee et al., 2023), and DINO(Zhang et al., 2022), which primarily rely on image modality for detection. The Multi-modal group, on the other hand, incorporates methods like Hybrid(V)(Zhong et al., 2023), VSR(Zhang et al., 2021), GLAM(Wang et al., 2023) and VGT(Da et al., 2023), which combine visual information with textual modalities to enhance detection performance. The results indicate that we achieve an mAP of 82.4 based on Cas-cadeRCNN with DiT, which shows the competitiveness among the SOTA approaches. Furthermore, we discover that the accuracy of Multi-modal methods is generally higher than that of Vision-based layout analysis models, which also demonstrates the effectiveness of integrating Multi-modal features. The comparison results with other methods on PubLayNet, D⁴LA and DocBank are as shown in **Appendix A.2**.

Table 5: Comparsion with other methods on DocLayNet test set.

| Method | Backbone | Caption | Footnote | Formula | List-Item | Page-Footer | Page-Header | Picture | Section-Header | Table | Text | Title | mAP |
|---|---|---|---|---|---|---|---|---|---|---|---|---|---|
| *Vision-based:* | | | | | | | | | | | | | |
| FasterRCNN(Ren et al., 2015) | ResNet50 | 69.9 | 70.5 | 59.3 | 80.7 | 61.6 | 70.8 | 70.3 | 66.8 | 81.3 | 84.2 | 74.8 | 71.8 |
| FasterRCNN(Ren et al., 2015) | ResNet101 | 70.1 | 73.7 | 63.5 | 81.0 | 58.9 | 72.0 | 72.0 | 68.4 | 82.2 | 85.4 | 79.9 | 73.4 |
| MaskRCNN(He et al., 2017) | ResNet50 | 68.4 | 70.9 | 60.1 | 81.2 | 61.6 | 71.9 | 71.7 | 67.6 | 82.2 | 84.6 | 76.6 | 72.4 |
| MaskRCNN(He et al., 2017) | ResNet101 | 71.5 | 71.8 | 63.4 | 80.8 | 49.3 | 70.0 | 72.7 | 69.3 | 82.9 | 85.8 | 80.4 | 73.5 |
| Yolov5x6(Pfitzmann et al., 2022) | - | 77.7 | 77.2 | 66.2 | 86.2 | 61.1 | 67.9 | 77.1 | 74.6 | 86.3 | 88.1 | 82.7 | 76.8 |
| CascadeRCNN(Cai & Vasconcelos, 2018) | DiT | 77.0 | 79.0 | 70.6 | 84.8 | 64.1 | 74.9 | 77.7 | 72.8 | 87.8 | 88.2 | 84.0 | 78.2 |
| TransDLANet(Cheng et al., 2023) | ResNet101 | 68.2 | 74.7 | 61.6 | 81.0 | 54.8 | 68.2 | 68.5 | 69.8 | 82.4 | 83.8 | 81.8 | 72.3 |
| SwinDocSegmenter(Banerjee et al., 2023) | Swin-Large | 83.6 | 64.8 | 62.3 | 82.3 | 65.1 | 66.4 | 84.7 | 66.5 | 87.4 | 88.2 | 63.3 | 76.9 |
| DINO(Zhang et al., 2022) | ResNet50 | 85.5 | 69.2 | 63.8 | 80.9 | 54.2 | 63.7 | 84.1 | 64.3 | 85.7 | 83.3 | 82.8 | 74.3 |
| DINO(Zhang et al., 2022) | ResNet101 | 71.8 | 78.8 | 72.7 | 85.6 | 63.0 | 76.6 | 74.1 | 72.1 | 87.3 | 87.6 | 85.1 | 77.7 |
| *Multi-modal:* | | | | | | | | | | | | | |
| Hybrid(V)(Zhong et al., 2023) | ResNet50 | 83.2 | 67.8 | 63.9 | 86.9 | 89.9 | 70.4 | 82.0 | 86.2 | 84.4 | 86.1 | 75.3 | 79.6 |
| Hybrid(V+BERT-3L)(Zhong et al., 2023) | ResNet50 | 81.9 | 68.5 | 64.2 | 87.5 | 86.1 | 81.3 | 81.8 | 84.2 | 85.4 | 85.6 | 82.3 | 80.8 |
| Hybrid(V+BERT-12L)(Zhong et al., 2023) | ResNet50 | 83.2 | 69.7 | 63.4 | 88.6 | 90.0 | 76.3 | 81.6 | 83.2 | 84.8 | 84.8 | 84.9 | 81.0 |
| VSR(Zhang et al., 2021) | ResNet101 | 72.6 | 72.1 | 73.8 | 86.2 | 81.8 | 81.3 | 63.1 | 82.4 | 79.4 | 88.4 | 80.7 | 78.4 |
| GLAM(Wang et al., 2023) | - | 74.3 | 72.0 | 66.6 | 76.7 | 86.2 | 78.0 | 5.7 | 79.8 | 56.3 | 74.3 | 85.1 | 68.6 |
| GLAM+YOLOv5x6(Wang et al., 2023) | - | 77.7 | 77.2 | 66.6 | 86.2 | 86.2 | 78.0 | 77.1 | 79.8 | 86.3 | 88.1 | 85.1 | 80.8 |
| VGT(Da et al., 2023) | DiT | 79.8 | 80.5 | 81.5 | 86.2 | 68.3 | 79.3 | 78.2 | 76.3 | 87.7 | 89.3 | 85.4 | 81.2 |
| Ours(FasterRCNN) | ResNet50 | 74.5 | 71.1 | 71.8 | 82.9 | 68.6 | 74.5 | 71.9 | 73.8 | 81.5 | 86.0 | 76.6 | **75.8** |
| Ours(CascadeRCNN) | DiT | 78.3 | 79.5 | 84.4 | 87.8 | 72.7 | 79.4 | 78.4 | 78.8 | 88.4 | 90.1 | 87.6 | **82.4** |

**FLOPS vs. mAP comparison.** We present a detailed analysis comparing our approaches with the previous SOTA model VGT(Da et al., 2023), in terms of mAP and FLOPS, as depicted in Table 6. VGT utilizes two DiT-based backbones to independently extract semantic features from images and texts, which are subsequently fused for layout analysis within a CascadeRCNN framework. In contrast, we accomplish Multi-modal data fusion at the data pre-processing stage with only a single backbone to extract the combined Multi-modal features. A direct comparison of the results across three datasets reveals that we not only surpass VGT in accuracy but also operate with approximately half the FLOPS. This stark contrast highlights the efficiency of our methods, offering a more optimized solution for DLA tasks.

Table 6: Comparison with VGT in FLOPS and mAP

| Method | Backbone | FLOPS | mAP | | | |
| | | | DocLayNet | D4LA | PubLayNet | DocBank |
|---|---|---|---|---|---|---|
| VGT(Previous SOTA) | DiT | 1248.9 | 81.2 | 68.8 | 96.2 | 84.1 |
| Ours(CascadeRCNN) | DiT | **643.7** | **82.4** | **69.6** | **96.7** | **85.0** |

## 5 CONCLUSION

In this paper, we present a novel data pre-processing strategy specifically tailored for enhancing the recognition capabilities in document layout analysis. We provide a comprehensive analysis of the prevailing methodologies: Vision-based and Multi-modal models, and introduce our innovative contributions: Graphic Multi-modal Data Fusion and Spatial Geometric Transformation. Our approaches are integrated into the pre-processing phase, ensuring a minimal computational overhead above the original detection models. The experimental results not only substantiate the superiority with the highest mAP but also highlight its efficiency, achieving these results with a reduced count of FLOPS compared to the SOTA performance. Furthermore, we validate the versatility and adaptability of our methods across FasterRCNN and CascadeRCNN, showing the potential as a pluggable solution.

In the future, we envision the expansion of our approaches to encompass a broader array of document scenarios, including large-scale document models, table recognition, and key information extraction tasks.

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

## A APPENDIX

### A.1 DETAILED INTRODUCTION TO DATASETS

**DocLayNet** is a large, manually annotated document layout analysis dataset. This dataset contains 80863 PDF pages with 1107470 annotations from 6 different data sources, including *Financial, Scientific, Patents, Manuals, Laws, and Tenders*, representing a wide range of layout diversity. There are 11 distinct labels in DocLayNet, namely *Caption, Footnote, Formula, List-item, Page-footer, Page-header, Picture, Section-header, Table, Text, and Title*. The dataset is meticulously partitioned into a training set comprising 69,375 pages, a validation set of 6,489 pages, and a test set containing 4,999 pages.

**D$^4$LA** is a more diverse and detailed dataset for document layout analysis. This dataset comes from 12 types of documents, including *Budget, Email, Form, Invoice, Letter, Memo, News article, Presentation, Resume, Scientific publication, Scientific report, Specification*. There are 27 defined document layout categories in D$^4$LA. Although this dataset only has 11,092 pages, there are 146,846 labels, which reflects its detailed annotation and inherently challenging nature. The dataset is organized into a training set of 8,868 pages and a validation set of 2,224 pages.

**PubLayNet** is a large, automatically labeled dataset for document layout analysis. The source is a subset of PubMed Central Open Access and generates annotations by matching the PDF and XML formats of articles. It contains 358,353 pages with 5 categories: *Text, Title, Tale, Figure, and List*. The dataset is segmented into a substantial training subset with 335,703 pages, a validation subset of 11,245 pages, and a testing subset containing 11,405 pages.

**DocBank** is a new large-scale DLA dataset constructed using weakly supervised methods. The current DocBank dataset consists of a total of 500K document pages, of which 400K are used for training, 50K for validation, and 50K for testing. There are 12 categories in this dataset. including *Abstract, Author, Caption, Date, Equation, Figure, Footer, List, Paragraph, Reference, Section, Table, Title*.

**M⁶Doc** is a large-scale DLA dataset that is multi-format, multi-type, multi-layout, multi-language. The dataset includes 9080 document images with 74 different annotation labels, totaling 237,116 annotation instances.

## A.2  COMPARSION WITH OTHER METHODS ON M⁶DOC, PUBLAYNET, D⁴LA AND DOCBANK

To demonstrate the effectiveness of our approaches, we conduct experiments on PubLayNet, D⁴LA and DocBank based on CascadeRCNN with DiT backbone, as shown in Table 8, Table 9 and Table 10. We also conduct experiments on the M⁶Doc dataset combined with the M²Doc model, and the experimental results are shown in Table 7. The results show that we also achieve SOTA performance on multiple benchmarks.

Table 7: Comparsion with other methods on M6Doc.

| Method | Backbone | AP50 | AP75 | Recall | mAP |
|---|---|---|---|---|---|
| SOLOv2 | ResNet101 | 67.5 | 51.4 | 61.5 | 46.8 |
| FasterRCNN | ResNet101 | 67.8 | 57.2 | 57.2 | 49.0 |
| MaskRCNN | ResNet101 | 58.4 | 46.2 | 50.8 | 40.1 |
| Cascade MaskRCNN | ResNet101 | 70.5 | 62.9 | 62.1 | 54.4 |
| HTC | ResNet101 | 74.3 | 67.2 | 68.1 | 58.2 |
| SCNet | ResNet101 | 73.5 | 65.1 | 67.3 | 56.1 |
| Deformable DETR | ResNet101 | 76.8 | 63.4 | 75.2 | 57.2 |
| QueryInst | ResNet101 | 67.1 | 58.1 | 71.0 | 51.0 |
| ISTR | ResNet101 | 80.8 | 70.8 | 73.2 | 62.7 |
| TransDLANet | ResNet101 | 82.7 | 72.7 | 74.9 | 64.5 |
| VSR | ResNet101 | 76.2 | 68.8 | 66.4 | 59.9 |
| DINO | ResNet101 | 84.6 | 76.7 | 82.9 | 68.0 |
| M²Doc(Cascade MaskRCNN) | ResNet101 | 78.0 | 70.7 | 67.9 | 61.8 |
| M²Doc(DINO) | ResNet101 | 86.7 | 79.4 | 82.5 | 69.9 |
| Ours(M2Doc Cascade MaskRCNN) | ResNet101 | **80.0** | **75.2** | **69.5** | **63.4** |
| Ours(M2Doc DINO) | ResNet101 | **87.6** | **81.2** | **84.1** | **70.8** |

Table 8: Comparsion with other methods on PubLayNet validation set.

| Method | Backbone | *Figure* | *List* | *Table* | *Text* | *Title* | mAP |
|---|---|---|---|---|---|---|---|
| FasterRCNN(Ren et al., 2015) | ResNeXt101 | 93.7 | 88.3 | 95.4 | 91.0 | 82.6 | 90.2 |
| MaskRCNN(He et al., 2017) | ResNeXt101 | 94.9 | 88.6 | 96.0 | 91.6 | 84.0 | 91.0 |
| VSR(Zhang et al., 2021) | ResNet101 | 96.4 | 94.7 | 97.4 | 96.7 | 93.1 | 95.7 |
| TransDLANet(Cheng et al., 2023) | ResNet101 | 96.6 | 95.2 | 97.2 | 94.3 | 89.2 | 94.5 |
| LayoutLMV3(Huang et al., 2022) | DiT | 97.0 | 95.5 | 97.9 | 94.5 | 90.6 | 95.1 |
| CascadeRCNN(Cai & Vasconcelos, 2018) | DiT | 96.9 | 94.8 | 97.6 | 94.4 | 88.9 | 94.5 |
| GLAM(Wang et al., 2023) | - | 20.6 | 86.2 | 86.8 | 87.8 | 80.0 | 72.2 |
| DINO(Zhang et al., 2022) | ResNet50 | 97.3 | 96.0 | 98.0 | 94.9 | 91.4 | 95.5 |
| Hybrid(V)(Zhong et al., 2023) | ResNet50 | 97.4 | 96.4 | 98.1 | 97.0 | 92.8 | 96.3 |
| Hybrid(V+BERT-3L)(Zhong et al., 2023) | ResNet50 | 97.2 | 96.3 | 98.1 | 97.7 | 93.1 | 96.5 |
| Hybrid(V+BERT-12L)(Zhong et al., 2023) | ResNet50 | 97.2 | 96.4 | 98.2 | 97.4 | 93.5 | 96.5 |
| UniDoc(Gu et al., 2021) | ResNet50 | 96.4 | 93.7 | 97.3 | 93.9 | 88.5 | 93.9 |
| VGT(Da et al., 2023) | DiT | 97.1 | 96.8 | 98.1 | 95.0 | 93.9 | 96.2 |
| Ours(CascadeRCNN) | DiT | 97.7 | 96.8 | 98.6 | 95.9 | 94.6 | **96.7** |

## A.3  VISUALIZATION OF THE ROBUSTNESS OF OUR METHODS

We transform an original document to demonstrate the robustness of our proposed methods and the detection performance is as shown in Figure 5. Before the implementation of our proposed methods, the original FasterRCNN is susceptible to fluctuations and produces false detections subjecting to images with minor transformations. In contrast, the trained FasterRCNN with our improved data pre-processing exhibits more stable inference results and higher robustness.

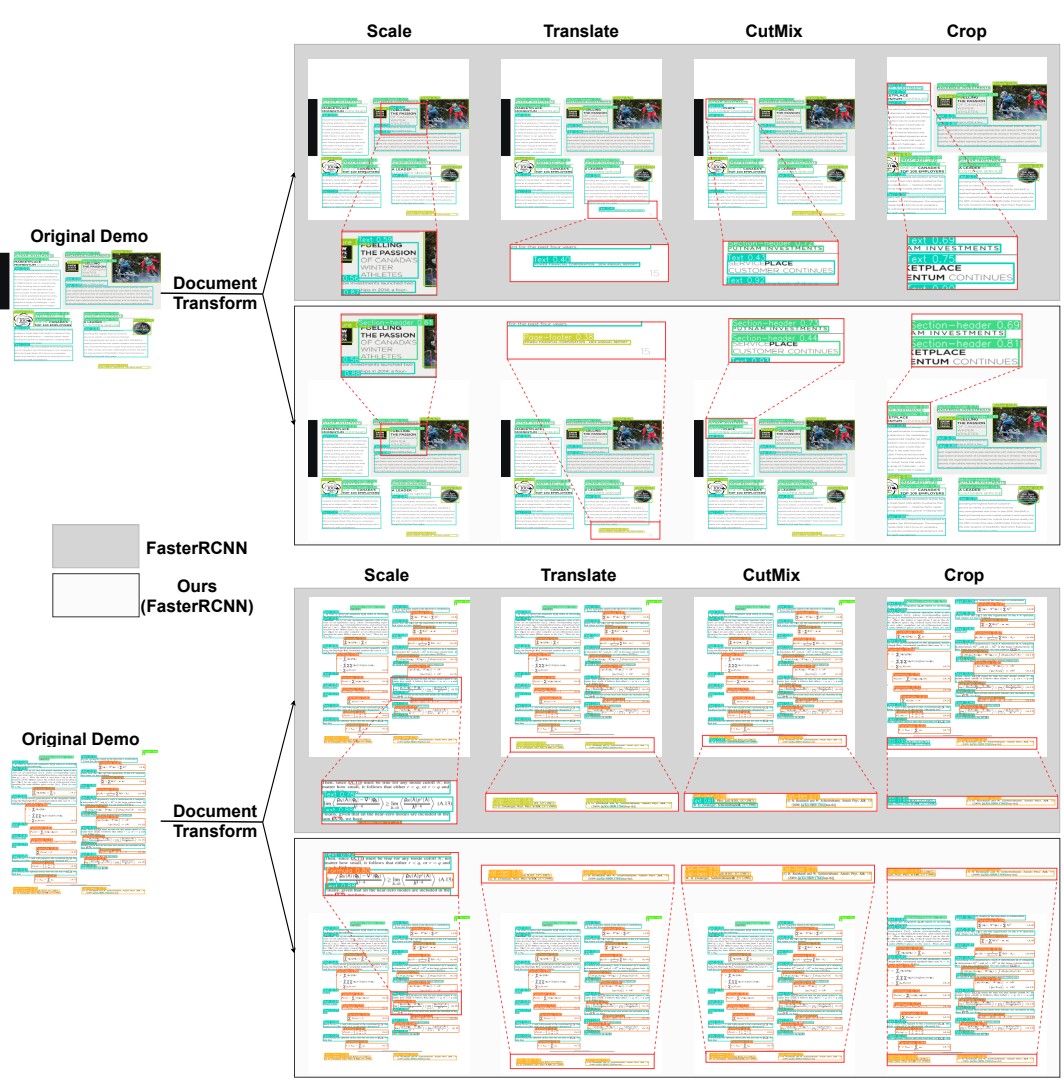

Figure 5: Comparsion of detection demo between original FasterRCNN and our proposed methods under document transform.

Table 9: Comparsion with other methods on D$^4$LA validation set.

| Method | Categories | | | | | | | | | mAP |
|--------|----------|--------|------------|----------|-----------|-----------|------------|--------|--------|-----|
| | *DocTitle* | *ListText* | *LetterHead* | *Question* | *RegionList* | *TableName* | *FigureName* | *Footer* | *Number* | |
| ResNeXt-101(Xie et al., 2017) | 70.6 | 71 | 82.8 | 48.4 | 76.1 | 66 | 45.9 | 76.2 | 83 | 65.1 |
| DiT-Base(Li et al., 2022) | 73.1 | 70.6 | 82.2 | 55 | 80.1 | 68.4 | 51.8 | 81.2 | 83.2 | 67.7 |
| LayoutLMV3(Huang et al., 2022) | 66.8 | 56.5 | 78.5 | 39.3 | 72.1 | 64.3 | 32.1 | 72.2 | 82.1 | 60.5 |
| VGT(Da et al., 2023) | 72.6 | 71.3 | 82.3 | 63.9 | 80.2 | 68.4 | 46.6 | 79.7 | 83.2 | 68.8 |
| Ours(CascadeRCNN) | 73.7 | 71.2 | 82.1 | 72.6 | 80 | 66.3 | 54.7 | 78.9 | 83.7 | **69.6** |
| | *ParaTitle* | *RegionTitle* | *LetterDear* | *OtherText* | *Abstract* | *Table* | *Equation* | *PageHeader* | *Catalog* | |
| ResNeXt-101(Xie et al., 2017) | 60.3 | 63.8 | 73.4 | 56.4 | 65.7 | 86.3 | 11.5 | 53.7 | 32 | 65.1 |
| DiT-Base(Li et al., 2022) | 63.2 | 67.5 | 74.5 | 59.2 | 73.8 | 86.2 | 9.2 | 56.5 | 44.8 | 67.7 |
| LayoutLMV3(Huang et al., 2022) | 55.6 | 59.5 | 70.8 | 50.8 | 68.2 | 80.6 | 7.3 | 53.1 | 37.3 | 60.5 |
| VGT(Da et al., 2023) | 63 | 67.2 | 76.7 | 60 | 80.4 | 86 | 19.9 | 56.6 | 40.9 | 68.8 |
| Ours(CascadeRCNN) | 62.8 | 67.5 | 74.3 | 60.7 | 69.6 | 86.9 | 32.4 | 55.8 | 48.9 | **69.6** |
| | *ParaText* | *Data* | *LetterSign* | *RegionKV* | *Author* | *Figure* | *Reference* | *PageFooter* | *PageNumber* | |
| ResNeXt-101(Xie et al., 2017) | 85.2 | 68.4 | 69.3 | 68.2 | 62.6 | 76.7 | 83.4 | 62.2 | 57.9 | 65.1 |
| DiT-Base(Li et al., 2022) | 86.4 | 69.7 | 71.6 | 68.8 | 66 | 77.2 | 83.4 | 65.5 | 58.3 | 67.7 |
| LayoutLMV3(Huang et al., 2022) | 81.6 | 62.5 | 60.4 | 59.4 | 59.3 | 72.2 | 74.9 | 62.1 | 52.8 | 60.5 |
| VGT(Da et al., 2023) | 86.2 | 71.3 | 75.5 | 70.1 | 67.6 | 76.7 | 85.6 | 66.5 | 58.7 | 68.8 |
| Ours(CascadeRCNN) | 85.4 | 69.8 | 72.2 | 69.2 | 66.7 | 77.5 | 83.6 | 67.9 | 59.7 | **69.6** |

Table 10: Comparison performance of DocBank

| Models | *Abstract* | *Author* | *Caption* | *Date* | *Equation* | *Figure* | *Footer* | *List* | *Paragraph* | *Reference* | *Section* | *Table* | *Title* | **mAP** |
|--------|----------|--------|---------|------|----------|--------|--------|------|-----------|-----------|---------|-------|-------|-----|
| ResNeXt-101(Xie et al., 2017) | 89.7 | 72.6 | 82.3 | 69.7 | 76.4 | 73.6 | 78.2 | 78.3 | 66.2 | 81.7 | 75.9 | 77.3 | 84.1 | 77.4 |
| DiT-Base(Li et al., 2022) | 91.1 | 75.4 | 83.1 | 73.4 | 77.8 | 75.7 | 80.2 | 82.7 | 67.3 | 83.8 | 77.0 | 80.8 | 86.8 | 79.6 |
| LayoutLMv3-Base(Huang et al., 2022) | 90.5 | 73.6 | 81.2 | 73.5 | 76.0 | 74.4 | 78.1 | 80.7 | 65.8 | 82.8 | 76.6 | 78.6 | 86.3 | 78.3 |
| VGT(Da et al., 2023) | 92.4 | 79.9 | 88.8 | 79.1 | 86.7 | 76.6 | 84.8 | 88.6 | 75.8 | 85.6 | 81.5 | 83.9 | 89.8 | 84.1 |
| **Ours** | **92.8** | **81.3** | **89.1** | **79.4** | **88.0** | **77.3** | **85.5** | **89.8** | **76.9** | **86.8** | **82.1** | **85.3** | **89.9** | **85.0** |

## A.4 DETAILED CODE FOR CONSTRUCTING EDGES IN SENTENCE GRAPH

The following code shows how the edges are established when we build the sentence graph. The core is to calculate the minimum distance as the reference and find the index of the nearest 4 nodes of each sentence. Since it is an undirected graph, we also consider the bidirectional structure.

```python
import numpy as np

def sentences_edges_construct(boxes, nearest_num=4):
    """
    Parameters
    ----------
    boxes: numpy.ndarray
    A numpy array of shape (N, 4), where each row represents a box
    with the format (x_min, y_min, width, height).

    nearest_num: int
    The number of nearest boxes to find for each box. Default is 4.

    Returns
    -------
    final_edge_index: numpy.ndarray
    An array of shape (2, E) containing the indices of nearest boxes.
    The first column contains the indices of the boxes, and the second
    column contains the indices of their nearest boxes.
    """

    # Calculate the x and y coordinates of the centers of the boxes
    centers_x = boxes[:, 0] + boxes[:, 2] / 2
    centers_y = boxes[:, 1] + boxes[:, 3] / 2

    # Calculate the absolute differences in x and y coordinates between
        the centers of all pairs of boxes
    dx = np.abs(centers_x[:, np.newaxis] - centers_x)
    dy = np.abs(centers_y[:, np.newaxis] - centers_y)

    # Calculate minimum distances along X and Y axes
    min_dx = (boxes[:, 2][np.newaxis,:] + boxes[:, 2][:, np.newaxis]) / 2
```

```
min_dy = (boxes[:, 3][np.newaxis,:] + boxes[:, 3][:, np.newaxis]) / 2

# Calculate the minimum distances between each pair of boxes
    considering the edge lengths
# Use a series of conditions to determine the minimum distance for
    each pair:
# 1. If the x-distance is less than the minimum possible x-distance,
    use the y-distance minus the minimum possible y-distance
# 2. If the y-distance is less than the minimum possible y-distance,
    use the x-distance minus the minimum possible x-distance
# 3. If both distances are greater than or equal to their respective
    minimums, use the Euclidean distance
# 4. If both distances are greater than their respective minimums,
    use 0 (no overlap)
min_dist = np.where((dx < min_dx) & (dy >= min_dy),dy - min_dy,
np.where((dx >=min_dx) & (dy < min_dy),dx - min_dx,
np.where((dx >= min_dx) & (dy >= min_dy),
np.sqrt((dx - min_dx) ** 2 + (dy - min_dy) ** 2),0)))

# Set the diagonal elements to infinity to ignore each rectangle's
    distance to itself
np.fill_diagonal(min_dist, np.inf)

# Find the indices of the nearest boxes for each rectangle
nearest_indices = np.argsort(min_dist, axis=1)[:, :nearest_num]

# Create an array of pairs of indices representing the connections
    between boxes and their nearest boxes
edge_index =
    np.column_stack((np.repeat(np.arange(len(nearest_indices)),
    nearest_num), nearest_indices.ravel())).T

# Swap the order of indices to create the reverse connections and
    concatenate them with the original pairs
new_edge_index = np.stack((edge_index[1], edge_index[0]), axis=1).T
final_edge_index = np.concatenate((edge_index,new_edge_index),axis=1)

# Remove duplicate pairs and return the final array of unique
    connections
final_edge_index = np.unique(final_edge_index, axis=1,
    return_index=False)

return final_edge_index
```

## A.5 VISUALIZATION OF THE ENTIRE PARSING PROCESS

Document layout analysis is a fundamental ability for structuring document content, and the entire parsing process includes additional steps such as Text box matching and Formatted output, as shown in Figure 6.

For the parsing of a document, the initial step involves converting it into an image format. Subsequently, we employ DLA model to discern and identify the various elements to facilitate the extraction of meaningful information.

The subsequent step of our methodology focuses on the precise alignment of detected boxes with the sentence boxes. This is achieved through the computation of Intersection over Union (IoU) scores for each line, which serves as a measure of the spatial overlap between the predicted and actual sentence boxes.

The final stage of our process entails the conversion of the content into a structured, hierarchical format. This structured output is designed to reflect the inherent organization, with a clear delineation of different levels of information. We need to give special attention to the ordering of content in multi-column PDF documents, as well as the hierarchical structure of headings. It is essential

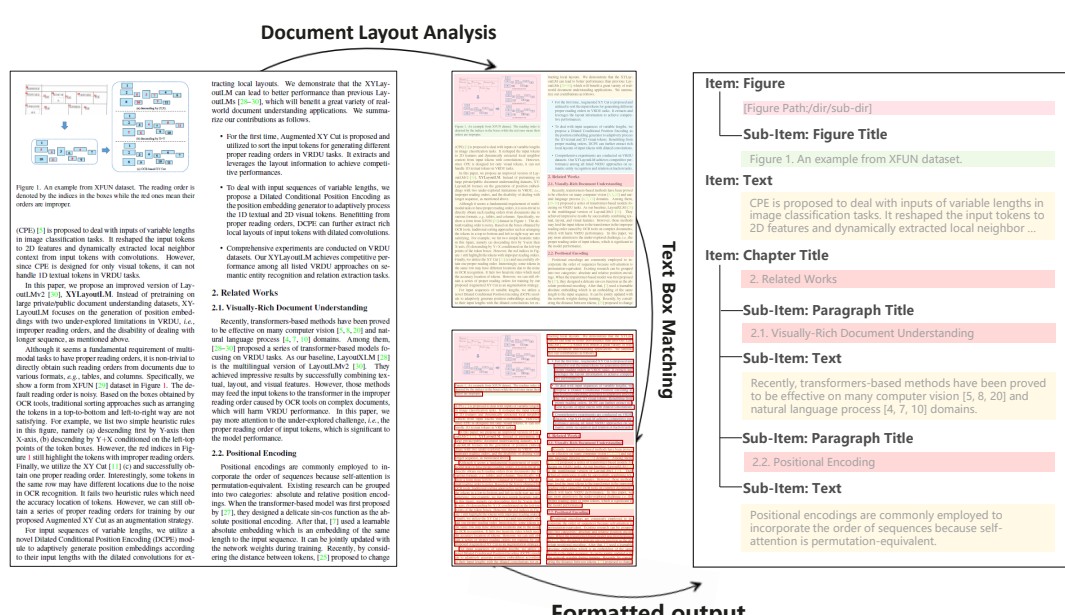

Figure 6: The entire document parsing process.

to maintain the correct sequence of output, especially in documents with complex layouts. For instance, in a multi-column PDF, the output order must accurately reflect the document structure. As illustrated in Figure 6, the section titled "2. Related Works" is correctly followed by its sub-sections, "2.1 Visually-Rich Document Understanding" and "2.2 Positional Encoding", demonstrating the importance of preserving the hierarchical structure for the output.

## A.6 DIFFERENCE BETWEEN OBJECT DETECTION AND DOCUMENT LAYOUT ANALYSIS

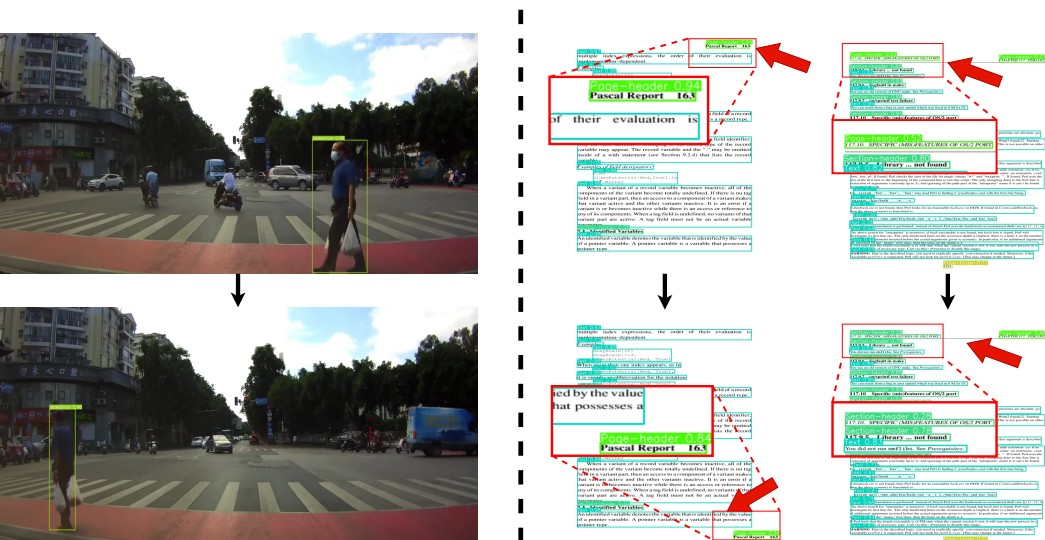

Figure 7: Difference between object detection and DLA. The left half is a typical pedestrian detection task, while the right half is a layout analysis task.

Compared with object detection, the difference in DLA is as shown in Figure 7. While textual semantic features do exist within documents, the relative positioning of sentences is more critical to determine the categories. In the case of object detection, regardless of how the pedestrian moves, the category always is pedestrian. On the contrary, if we move a "Page-header" in a document to the position of "Page-footer", it is unreasonable to still recognize it as a "Page-header". This phenomenon implies that DLA task lacks translation equivariance. Secondly, due to the enormous diversity of document structures, the robustness of DLA models needs to be improved. As shown in the third column of Figure 7, we randomly shift the position of "Page-header" slightly, and its category changes to "Section-header". DLA models must learn to recognize and adapt to the subtle differences of structure to accurately analyze the different elements in the document, rather than relying solely on the semantic content of texts.

With the different characteristics, that is why we propose Graphic Multi-modal Data Fusion and Spatial Geometric Transformation in the data pre-processing stage to improve the performance of layout analysis tasks.

