# OpenReview forum: "Boosting Document Layout Analysis with Graphic Multi-modal Data Fusion and Spatial Geometric Transformation"
_ICLR.cc/2025/Conference — Submitted to ICLR 2025_

### Official Review · Reviewer_PMaH · 2024-10-29

**Soundness:** 3
**Presentation:** 3
**Contribution:** 3
**Rating:** 6
**Confidence:** 4

**Summary:**

Current document layout analysis methods are mainly divided into two categories: one is based on computer vision technology, which focuses on image modality but ignores the textual modality in documents; the second is based on multimodal technology, integrating word embeddings to improve recognition accuracy, but inevitably increasing the computational burden after introducing textual modality. The authors found that the relative relationships among elements in the document affect the element categories, thus proposing Graphical Multi-modal Data Fusion technique, which is an image preprocessing module. Its main purpose is to construct a graph to establish connections between different disparate textual segments. At the same time, to enhance the robustness of the model, the authors designed Spatial Geometric Transformation strategy, which enhances the diversity of document structure in three dimensions: sentence, paragraph, and page. This strategy is also used in the image preprocessing stage. The authors skillfully leveraged "multimodal" information to improve accuracy without incurring much additional computation through their proposed strategies. Experimental results show that the authors' strategies have demonstrated state-of-the-art performance on multiple document layout analysis datasets.

**Strengths:**

Graphical Multi-modal Data Fusion technique proposed by the authors effectively integrates multimodal information, combining image modality and textual modality into a Fusion Image that is input into the model. This clever use of "multimodal" information significantly enhances the model's accuracy without substantially increasing its computational load. In addition to this, Spatial Geometric Transformation strategy put forward by the authors introduces variations at the sentence, paragraph, and page levels to enhance the diversity of document structures. This not only improves the model's robustness but also further increases its accuracy.

**Weaknesses:**

1. The authors claim that the proposed method not only facilitates integration with existing models but also achieves significant accuracy improvements with negligible extra computations. However, the author does not discuss why such methods can "negligible extra computations"; They only conducted ablation experiments to prove that their methods do not bring additional burdens.

2. There is a lack of experiments on other commonly used layout analysis datasets, such as PubLayNet and M6Doc. Furthermore, there is no comparison with the latest methods, such as M2Doc. Whether the proposed method can be applied to or integrated with the most recent approaches, such as M2Doc. Additionally, the performance comparisons and ablation studies are only conducted on DocLayNet, which is not convincing.

[1] Zhong X, Tang J, Yepes A J. Publaynet: largest dataset ever for document layout analysis[C]//2019 International conference on document analysis and recognition (ICDAR). IEEE, 2019: 1015-1022.

[2]Cheng H, Zhang P, Wu S, et al. M6doc: A large-scale multi-format, multi-type, multi-layout, multi-language, multi-annotation category dataset for modern document layout analysis[C]//Proceedings of the IEEE/CVF Conference on Computer Vision and Pattern Recognition. 2023: 15138-15147.

[3]Zhang N, Cheng H, Chen J, et al. M2Doc: A Multi-Modal Fusion Approach for Document Layout Analysis[C]//Proceedings of the AAAI Conference on Artificial Intelligence. 2024, 38(7): 7233-7241.

**Questions:**

In the paper, it would be advantageous to highlight the performance improvements achieved with minimal additional computational cost, along with an analysis that explains the underlying reasons for these gains. This approach would not only more effectively underscore the key contributions of the work but also pave the way for future research in this area. Furthermore, for aesthetic purposes, it is suggested to shift the 'Parser / OCR' label in Figure 2(a) slightly to the left. To strengthen the credibility of the proposed method, it is advisable to conduct a comparative analysis against previous methods on more benchmarks, as well as to include ablation studies on additional datasets.

---

> ### Author Response · Authors · 2024-11-18
>
> **We would like to express our sincere thanks for the constructive comments and suggestions from the reviewer.**
>
> **Weak1: The authors claim that the proposed method not only facilitates integration with existing models but also achieves significant accuracy improvements with negligible extra computations. However, the author does not discuss why such methods can "negligible extra computations"; They only conducted ablation experiments to prove that their methods do not bring additional burdens.**
>
> A-Weak1: The methods we proposed in this paper involves data preprocessing, which include GMDF with a small number of parameters and SGT without parameters. The former intervenes in the model inference process, while the latter only exists in the training process.
>
> After obtaining the 768 dimensional embeddings for each token, we create a linear layer to map them into 3 dimensions to reduce computational complexity. So whether wordgraph or sentential graph, the multimodal information of each node is 6-dimensional. Therefore, even if there are 1000 tokens in one page of the document, the additional computational burden caused by GMDF does not exceed 1.5 GFlops. For comparsion, the flops of basic model of CascadeRCNN-DiT is 642.2 GFlops, so the increased computations is negligible.
>
> In terms of the information loss caused by mapping 768 dimensional token embeddings to 3 dimensions, we conduct experiments based on Faster RCNN and DocLayNet. The results are as follows:
>
> | Method       | mAP  | FLOPS(G) |
> | ------------ | ---- | -------- |
> | FasterRCNN   | 71.8 | 134.4    |
> | GMDF-3 dim   | 75.2 | 135.9    |
> | GMDF-64 dim  | 75.2 | 166.1    |
> | GMDF-128 dim | 75.3 | 197.6    |
>
> With the increase of dimension, the mAP has hardly improved, while the FLOPS has increased significantly, even the Flops of 128 dim is 1.5 times that of base model. **Therefore, it's to use graphs and attention mechanism to construct the connections between words and sentences, the dim for data fusion does not need to be very large, as the backbone will extract high-level features based on the connections. **
>
>
> **Weak2: There is a lack of experiments on other commonly used layout analysis datasets, such as PubLayNet and M6Doc. Furthermore, there is no comparison with the latest methods, such as M2Doc. Whether the proposed method can be applied to or integrated with the most recent approaches, such as M2Doc. Additionally, the performance comparisons and ablation studies are only conducted on DocLayNet, which is not convincing.**
>
>
> A-Weak2: We have demonstrated the effectiveness of our method with multiple DLA benchmarks. The comparative experiments are not only conducted on DocLayNet, but also on other PublayNet, DocBank, and D4LA. The results are shown in Tab.6, Tab.7 and Tab.8 at the Appendix part.
>
> In addition, we supplement experiments based on the M6Doc dataset with M2Doc methods, as shown below. It can be seen that our method can still improve the accuracy of the M2Doc model.
>
> | Method       |Backbone| mAP  |
> | ------------ | ---- | ---- |
> | M2Doc(Cascade Mask-RCNN)  | ResNet-101 | 61.8 |
> | M2Doc(Cascade Mask-RCNN) + GMDF  | ResNet-101 | 63.4 |
> | M2Doc(DINO)  | ResNet-101 | 69.9 |
> | M2Doc(DINO) + GMDF  | ResNet-101 | 70.8 |
>
> **Q1: Furthermore, for aesthetic purposes, it is suggested to shift the 'Parser / OCR' label in Figure 2(a) slightly to the left**
>
> A-Q1: Thanks for the suggestion, we will revise that in the final version of this paper.
>
>
> **Again, thank you very much for the kind efforts in evaluating and helping to improve the quality of our manuscript!**

---

> ### Author Response · Authors · 2024-11-22
>
> The modified contents have been presented in the full paper. If you have any other questions, please feel free to contact us at any time.
>
> Thank you!

---

> > ### Comment · Reviewer_PMaH · 2024-11-25
> >
> > Thanks for the authors' response. Most of my concerns have been addressed, and I would like to raise my rating.

---

> > > ### Author Response · Authors · 2024-11-25
> > >
> > > Thank you again for your valuable feedback that has helped us improve the quality of this paper!

---

> > > > ### Author Response · Authors · 2024-12-02
> > > > **A friendly reminder on re-evaluating the value of this paper**
> > > >
> > > > Dear Reviewer,
> > > >
> > > > We sincerely appreciate the discussion, which is vital to enhance the quality and persuasiveness of the paper.
> > > >
> > > > Based on our previous discussion, the main revisions to the original paper are as follows:
> > > >
> > > > 1. We have analyzed that our method does not bring computational load mainly because the information dimension of GMDF is relatively low. Besides, we have conducted an ablation experiment on the textual dimension of GMDF. We find that there is almost no difference in accuracy when projecting 768 dimensions to 3 dimensions and 128 dimensions, but 3 dimensions bring almost no computational burden. This confirms our previous intention: it's to use graphs and attention mechanism to construct the connections between words and sentences, the dim for data fusion does not need to be very large, because the backbone will extract high-level features based on the connections.
> > > >
> > > > 2. We have supplemented the experiment of the joint M2Doc model on the M6Doc dataset, and the experimental results showed that our method is universal.
> > > >
> > > > 3. We have modified the position of 'Parser / OCR' label in Figure 2(a) to make it more aesthetic.
> > > >
> > > > **From our previous discussions, it seems that our revisions and explanations have resolved your questions. As the deadline for discussion is approaching, could you take a look at the updated paper, and re-evaluate its value?**
> > > >
> > > > If you have any follow-up concerns, please let us know and we will be happy to answer them.
> > > >
> > > > Best regards,
> > > >
> > > > the Authors

---

### Official Review · Reviewer_eYjB · 2024-10-31

**Soundness:** 2
**Presentation:** 1
**Contribution:** 2
**Rating:** 5
**Confidence:** 4

**Summary:**

The manuscript proposes two methods in the pre-processing stage of a document layout analysis (DLA) task. The proposed models fall under the category of multi-modal methods, as they combine semantic information (e.g. word embeddings) along with the vision-based information (e.g. spatial positions). While a few multi-modal methods exist, the primary contribution of the paper is on computational cost savings, while still achieving a better mAP score.

The first method is a Graphic Multi-modal data fusion (GMDF) stage, which constructs a graph from spatial and semantic relationships. The original document image is parsed through an OCR, and tokenized. The word embeddings are used to from word and sentence graphs. The fusion of the spatial and semantic graph in then fed to the backbone of the network.

The second method is a data augmentation technique. Various augmentations are produced based on sentence remixing, paragraph perturbation and crops are used to improve the generalization of the model.

The approach is evaluated on 4 DLA datasets (DocLayNet, D4LA, PubLayNet and Docbank), and against 2 detectors: FasterRCNN and CascadeRCNN. Compared to VGT (previous SOTA), the average mAP score goes up by 1.0 while reducing the FLOPS to half.

**Strengths:**

The paper's has a few strengths in significance: computational cost, pluggable pre-processing step and a multi-modal approach to gather both semantic and spatial information. The proposed GMDF + SGT stages are able to perform as good as SOTA VGT method with nearly half the FLOPs. The proposed approaches are a pre-processing step, whose output is fed to the backbone of any DLA network. This makes the method applicable to a lot more use-cases. Lastly, using both textual and spatial information usually outweighs vision-only approach, and this method proves this again.

In terms of experiments, ablation studies are performed, along with comparisons with both vision and multi-modal approaches. The comparison with VGT method is clear in terms of mAP and FLOPs.

**Weaknesses:**

The primary weaknesses of the paper are insufficient experiments, reasoning of success, and originality.

Originality: It is unclear how a text and spatial graph is novel. The paper itself talks about SOTA approaches that have done the same, but is not able to clarify what makes the GMDF stage novel. Creating a word graph, and constructing a sentence graph from it has been a known and well studied work. As graph-based methods have known scaling challenges, it is unclear why the given approach will flare better? The second stage SGT is a well-known data augmentation technique. It does leave open questions like -
- What is the effect of scaling on OCR parsing?
- Why augment here instead of abstracting it away from the architecture, and pulling it in training procedures

Reasoning of Success: On the lines of novelty, it is unclear why does the approach perform better? What are the areas it fails and why?

Experiments: Lastly, while the mAP scores are compared against a gamut of SOTA approaches, VGT is picked as the only one from the list for FLOPs comparison. How does the proposed approach compares to other SOTA techniques with similar mAP scores: Hybrid (V+BERT-3L), Hybrid (V+BERT-12L), GLAM+YOLOv5x6?
- How does the approach generalize to non-standard documents with unstructured layouts?
- How does it get affected by a bad OCR result? Do the semantic information start to hurt more than help then?
The data augmentations are also limited to scaling, translating and cropping. It does not seem to capture the real world issues like skewing, font changes, noise artifacts etc.

**Questions:**

- As graph-based methods have known scaling challenges, it is unclear why the given approach will flare better? The second stage SGT is a well-known data augmentation technique. It does leave open questions like -
- What is the effect of scaling on OCR parsing?
- Why augment here instead of abstracting it away from the architecture, and pulling it in training procedures
- why does the approach perform better? What are the areas it fails and why?
- How does the proposed approach compares to other SOTA techniques with similar mAP scores: Hybrid (V+BERT-3L), Hybrid (V+BERT-12L), GLAM+YOLOv5x6?
- How does the approach generalize to non-standard documents with unstructured layouts?
- How does it get affected by a bad OCR result? Do the semantic information start to hurt more than help then?

---

> ### Author Response · Authors · 2024-11-18
>
> **We would like to express our sincere thanks for the constructive comments and suggestions from the reviewer.**
>
> **Weak1: It is unclear how a text and spatial graph is novel. The paper itself talks about SOTA approaches that have done the same, but is not able to clarify what makes the GMDF stage novel. Creating a word graph, and constructing a sentence graph from it has been a known and well studied work. As graph-based methods have known scaling challenges, it is unclear why the given approach will flare better?**
>
> A-Weak1: Our innovation lies in the multimodal data fusion of text regions in document images based on graph structures, which is different from previous image-based layout analysis models. The core of the idea is the construction of two graph structures, Wordgraph for extracting internal information and Sentencegraph for outer representation.
>
> In fact, layout analysis tasks are similar to object detection in computer vision, essentially based on a document image, detecting the categories and coordinates of elements in the image. But an important difference is that documents are rich in textual information, which is useful for layout analysis. Therefore, existing SOTA methods, such as VGT and VSR, consider both image and text features simultaneously. However, the drawback of those multimodal methods is their excessive computational load. As shown in Fig.2 (b), previous multimodal practices, essentially follow a multimodal feature fusion approach, treating text information as a grid map. This information is processed through a separate backbone to extract deep-level features before fusion, which leads to a multiplicative increase in computational requirements.
>
> Considering these issues and analyzing that the categories are largely determined by relative relationships, we construct a graphic method for multimodal fusion. Moreover, to prevent the computational load from becoming too large, our approach first performs multimodal fusion based on images and establishes associations, then extracts features with backbone. This has two advantages: compared to previous multimodal layout analysis models, our method introduces a minimal increase in computational load yet surpassing their mAP on multiple Benchmarks. Additionally, since our method integrates features during the pre-processing stage, it can be combined with multiple models without altering the original model architecture, making it a pluggable solution for application.
>
> As shown in Fig.2, our method is a new framework, which is different with previous Vision-based and Muti-modal. To the best of our known, there is currently no method for multimodal data fusion with "Wordgraph in Sentencegraph" in DLA tasks.
>
>
> **Weak2: The second stage SGT is a well-known data augmentation technique. It does leave open questions like**
> **–What is the effect of scaling on OCR parsing?**
> **-Why augment here instead of abstracting it away from the architecture, and pulling it in training procedures**
>
> A-Weak2: We first use OCR to recognize characters and obtain the coordinates of the text based on the original image. When we scale the document image, the characters remain unchanged and only the corresponding coordinates need to be transformed. Therefore, scaling does not affect the OCR results.
>
> For the other question, since SGT is to increase the diversity of training data, we cannot add it to the inference process to ensure the stability of predictions. GMDF is a multimodal data fusion method with trainable parameters and therefore needs to be included in the inference process. The proposal of SGT focuses more on improving the robustness of models in the application process. Because the real layout analysis scenario cannot be as clean as the open-source research benchmark. For a Word document, it is unavoidable to make subtle transformations to the position of text blocks, or the size of an image.  As shown in Fig.6 in the Appendix, there are prediction errors due to random slight movement of some text positions. Therefore, we propose using the SGT method to simulate the subtle change of documents to improve the generalization during the training process.
>
> Different from most existing image data augmentation schemes that rely on whole images, our proposed SGT utilizes a hierarchical strategy that divides into three different dimensions: replacing text at sentence-level, scaling and translating at paragraph-level, and cropping at page-level.
>
> In addition, this method does not introduce any parameters yet enhances the data diversity during the training process. It is effective in improving the robustness of the model and is more beneficial for practical application.

---

> ### Author Response · Authors · 2024-11-18
>
> **Weak3: why does the approach perform better? What are the areas it fails and why?**
>
> A-Weak3: Our method is effective for two main reasons: it establishes connections between sentences and utilizes multimodal information. First, in the layout analysis task, we find that the relative relationships between texts determine their categories. For example, we consider a sentence to be a "section-Header" because it possess a thicker font, larger font size, and shorter length compared to regular text. Imagine this page with only this section header and no other elements, then it's hard to consider it still a section header. It is different with object detection. For example, we use a model to detect cats and dogs in an image. Even if all cats in the image are removed, the remaining objects can still be recognized as "Dog". This is the essential difference between layout analysis tasks and general object detection. Therefore, this is also why using graphs to construct connections in the preprocessing stage is effective for document layout analysis. The second reason is that documents possess both image attributes and rich textual attributes. We treat each word and sentence as a node, allowing us to combine the corresponding image and text information to represent this node. Finally, we fuse the association with the original image and utilize feature extraction module to amplify this implicit representation. The experimental results demonstrate the effectiveness of this method. It also implies that optimizing data preprocessing methods can achieve significant improvements in accuracy with minimal computational increases.
>
> When a document page contains only images and no text, GMDF is not working. If the images are relatively simple, the task is straightforward. However, if the images are highly complex and lack clear boundaries, it becomes challenging to accurately predict the bounding box. This is not just a problem we encounter, but a common challenge faced by current layout analysis models.
>
>
> **Weak4: Lastly, while the mAP scores are compared against a gamut of SOTA approaches, VGT is picked as the only one from the list for FLOPs comparison. How does the proposed approach compares to other SOTA techniques with similar mAP scores: Hybrid (V+BERT-3L), Hybrid (V+BERT-12L), GLAM+YOLOv5x6?**
>
> A-Weak4: Our method outperforms these methods in terms of accuracy. We can acquire the computational efficiency of recently open-sourced state-of-the-art (SOTA) model VGT, yet these models lack open-source code and do not consider computational costs in their papers, hence we cannot obtain an exact value of computational load. However, these models integrate multiple large-scale models, requiring coordination among various models, which leads to a more complex architecture and higher FLOPs. In contrast, ours is a single model and convenient to deploy. It is convincing that we achieve higher accuracy with almost half computations compared with the open-source SOTA model.
>
>
> **Weak5: How does the approach generalize to non-standard documents with unstructured layouts?**
>
> A-Weak5: The document datasets we study are not all in standard paper format. There are a large number of non-standard format documents in the D4LA and DocLayNet datasets, such as Email, Letter, Resume, Memo, Specification, Magazine, etc. Non-standard documents are essentially a collection of visual and textual information, so layout analysis of such documents is consistent with that of stand documents. There is no essential difference. Our method can improve the accuracy of the model on D4LA and DocLayNet, so it is also applicable.
>
> **Weak6: How does it get affected by a bad OCR result? Do the semantic information start to hurt more than help then?**
>
> A-Weak6: In fact, OCR is currently a very mature technology, and its accuracy in document recognition can generally reach over 99%. Since our method utilizes documents to obtain OCR results and then trains with this information, the training process is accompanied by correct or little incorrect OCR results. Even if an occasional character prediction error occurs, it has little impact on the OCR results and can be corrected through training. In addition, the robustness of our model is strong, and little OCR recognition errors have almost no impact on our layout analysis performance.
>
> **Again, thank you very much for the kind efforts in evaluating and helping to improve the quality of our manuscript!**

---

> ### Author Response · Authors · 2024-11-22
>
> The modified contents have been presented in the full paper. If you have any other questions, please feel free to contact us at any time.
>
> Thank you!

---

> > ### Author Response · Authors · 2024-12-02
> > **A friendly reminder on re-evaluating the value of this paper**
> >
> > Dear Reviewer,
> >
> > We sincerely appreciate the discussion, which is vital to enhance the quality and persuasiveness of the paper.
> >
> > Based on the upper questions, our explanation can be summarized as follows:
> >
> > 1. Our innovation lies in the multimodal data fusion of text regions in document images based on graph structures, which is different from previous image-based layout analysis models.
> >
> > 2. We first use OCR to recognize characters and obtain the coordinates of the text based on the original image. When we scale the document image, the characters remain unchanged and only the corresponding coordinates need to be transformed. Therefore, scaling does not affect the OCR results. SGT is used to increase the diversity of training data and reduce overfitting. In machine learning, it cannot be included in the inference process.
> >
> > 3. Our method is effective because we establish connections between sentences and utilizes multimodal information. When a document page contains only images and no text, GMDF is not working. If the images lack clear boundaries, it becomes challenging to accurately predict the bounding box. This is not just a problem we encounter, but a common challenge faced by current layout analysis models.
> >
> > 4. VGT is the previous SOTA model, so we mainly compared with it, and the VGT code is also open source. Other models are not open source and the computational complexity is not considered in the paper, so we cannot obtain an accurate FLOPs. However, these models integrate multiple large-scale models, requiring coordination among various models, which leads to a more complex architecture and higher FLOPs.
> >
> > 5. There are a large number of non-standard format documents in our research datasets D4LA and DocLayNet, such as Email, Letter, Resume, Memo, Specification, Magazine, etc. Non-standard documents are essentially a collection of visual and textual information, so layout analysis of such documents is consistent with that of stand documents. There is no essential difference. Our method can improve the accuracy of the model on D4LA and DocLayNet, so it is also applicable.
> >
> > 6. OCR is currently a very mature technology, and its accuracy in document recognition can generally reach over 99%. Due to the possibility of minimal OCR errors during the training process, if OCR errors occur, there is almost no impact during the inference process.  In addition, the robustness of our model is strong, and little OCR recognition errors have almost no impact on our layout analysis performance.
> >
> > **As the deadline for discussion is approaching, could you take a look at the updated paper, and re-evaluate its value?**
> >
> > If you have any follow-up concerns, please let us know and we will be happy to answer them.
> >
> > Best regards,
> >
> > the Authors

---

### Official Review · Reviewer_ssuv · 2024-10-31

**Soundness:** 2
**Presentation:** 3
**Contribution:** 2
**Rating:** 5
**Confidence:** 4

**Summary:**

The paper describes a simple method for image pre-processing specially designed to obtain a better image representation for document layout analysis. Two different pre-processing strategies are proposed: on one hand, enriching image information with semantic information obtained through word embedding and the relations between words and sentences in the document. On the other hand, a data augmentation strategy specific for document layout analysis. The proposed strategies can be integrated with several existing DLA methods with a very reduced extra computation cost. Experimental results analyze the contribution of the two proposed pre-processing strategies and compare the performance with current SoA on standard benchmarks, showing SoA performance.

**Strengths:**

- The proposed method is simple, efficient and can be integrated into several DLA methods.
- Experimental results show that the Graphical Multi-modal Data Fusion module improves the results of Faster-RCNN and Cascade-RCNN, obtaining SoA results on the standard benchmarks.

**Weaknesses:**

- Although the experiments show an improvement in the performance when the pre-processing step is applied, it is not clear to me which is the reason that brings that improvement, since the proposed fusion approach does not seem very intutitive to me. It is just the result of summing raw pixel values with some very highly compressed semantic word and sentence information, which can be very unrelated values. It would be valuable if you could explain the rationale behind the fusion of raw pixel values with semantic information.Moreover, semantic word embedding undergoes an aggressive projection from 768 dimensions to only 3, with the risk of high informaton loss. It is difficult to me to visualize which enriched information is added to the original image in eq. (8). Some discussion on this would be useful.
- The ablation analysis is incomplete. The contribtuion of the two strategies is evaluated but the multimodal fusion module should be analyzed with a deeper detail. It should be analyzed the contribution of word and sentence graphs (i.e, what are the results using only the word graph or the sentence graph?). The contribution of the attention on the graph should also be analyzed (comparing results using the sentence/word features with and without attention on the graph) since one of the starting hypothesis is that the relations among elemetns in the document are relevant for DLA.
- The contribution of the SGT is a bit marginal

**Questions:**

- The fusion strategy only modifies the original image where there is text. What about non-text elements, such as tables or figures? What is the impact of the method on those regions?
- In equation (8), is the token embedding w_m the result of attention on word graph? Or is it just the original token embedding without attention?

---

> ### Author Response · Authors · 2024-11-18
>
> **We would like to express our sincere thanks for the constructive comments and suggestions from the reviewer.**
>
> **Weak1: Although the experiments show an improvement in the performance when the pre-processing step is applied, it is not clear to me which is the reason that brings that improvement, since the proposed fusion approach does not seem very intutitive to me. It is just the result of summing raw pixel values with some very highly compressed semantic word and sentence information, which can be very unrelated values. It would be valuable if you could explain the rationale behind the fusion of raw pixel values with semantic information.Moreover, semantic word embedding undergoes an aggressive projection from 768 dimensions to only 3, with the risk of high informaton loss. It is difficult to me to visualize which enriched information is added to the original image in eq. (8). Some discussion on this would be useful.**
>
> **A-Weak1:**
>
> 1. Our method is effective for two main reasons: it establishes connections between sentences and utilizes multimodal information. First, in the layout analysis task, we find that the relative relationships between texts determine their categories. For example, we consider a sentence to be a "section-Header" because it possess a thicker font, larger font size, and shorter length compared to regular text. Imagine this page with only this section header and no other elements, then it's hard to consider it still a section header. It is different with object detection. For example, we use a model to detect cats and dogs in an image. Even if all cats in the image are removed, the remaining objects can still be recognized as "Dog". This is the essential difference between layout analysis tasks and general object detection. Therefore, this is also why using graphs to construct connections in the preprocessing stage is effective for document layout analysis. **By the way, the relation is not the association between images and texts, but the association between sentences and words, with node features that combine image and textual information.**
>
> The second reason is that documents possess both image attributes and rich textual attributes. We treat each word and sentence as a node, allowing us to combine the corresponding image and text information to represent this node. Finally, we fuse the association with the original image and utilize feature extraction module to amplify this implicit representation. The experimental results demonstrate the effectiveness of this method. It also implies that optimizing data preprocessing methods can achieve significant improvements in accuracy with minimal computational increases.
>
> 2. Semantic word embedding undergoes an aggressive projection from 768 dimensions to only 3 is to reduce computations. For the information loss, we conduct experiments based on FasterRCNN on DocLayNet and increase the feature dimensions to 64 and 128. The results are shown below:
>
> | Method       | mAP  | FLOPS(G) |
> | ------------ | ---- | -------- |
> | FasterRCNN   | 71.8 | 134.4    |
> | GMDF-3 dim   | 75.2 | 135.9    |
> | GMDF-64 dim  | 75.2 | 166.1    |
> | GMDF-128 dim | 75.3 | 197.6    |
>
> The mAP has hardly improved, while the FLOPS has increased significantly, even the Flops of 128 dim is 1.5 times that of base model. **As a result, it's to use graphs and attention mechanism to construct the connections between words and sentences, the dim for data fusion does not need to be very large, because the backbone will extract high-level features based on the connections. **
>
> 3. eq.8 is to distinguish the blank areas (background) between words (foreground) inside a sentence. The fused information of each sentence areas can be acquired by GMDF. For the words region, we need to combine the 3-dim word embedding with the sentence information, as implied in the bottom line of Fig.3. For blank areas, only the sentence information is needed. If the word embedding line is removed, it will result in a partial loss of accuracy. For example, Cascade-RCNN reduces mAP by 0.4 if there is only GMDF without addition of word embedding.

---

> ### Author Response · Authors · 2024-11-18
>
> **Weak2: The ablation analysis is incomplete. The contribtuion of the two strategies is evaluated but the multimodal fusion module should be analyzed with a deeper detail. It should be analyzed the contribution of word and sentence graphs (i.e, what are the results using only the word graph or the sentence graph?). The contribution of the attention on the graph should also be analyzed (comparing results using the sentence/word features with and without attention on the graph) since one of the starting hypothesis is that the relations among elemetns in the document are relevant for DLA.**
>
> **A-Weak2:** Thanks for your suggestion. We have added ablation experiments based on the Cascade DiT model on DocLayNet.
>
> Firstly, wordgraph is a submodule of sentencegraph. When we directly take the average of the node features within each sentence without wordgraph, and take them as the node features of the sentence graph, the mAP is 81.3. After introducing wordgraph, the mAP improves to 82.1. In addition, if there is no graph attention mechanism, our method returns to directly adding word embedding to the original image, resulting in only 78.6 mAP, which is a small improvement compared to the baseline. As the response to the previous question, if we don't consider word embedding, which means removing the bottom line in Fig. 3, the mAP also decrease to 81.7 because there is no distinction between foreground and background areas within each sentence. The experimental results verifies that establishing sentence connections can effectively improve the layout analysis performance. Then, constructing a wordgraph can refine the features of sentences. Finally, using word embedding to distinguish foreground and background can further improve the accuracy of the model.
>
> | Detector     | Backbone  | WordEmbedding | SentenceGraph | WordGraph | mAP |
> | ------------ | --------- | --------      |--------       |--------   |-----|
> | Cascade-RCNN | DiT       |              |               |           |  78.0   |
> | Cascade-RCNN | DiT       |      Y         |               |           |  78.6   |
> | Cascade-RCNN | DiT       |      Y         |        Y       |           |  81.3   |
> | Cascade-RCNN | DiT       |      Y         |        Y       |     Y      |  82.1   |
> | Cascade-RCNN | DiT       |               |        Y       |     Y      |  81.7   |
>
> **Weak3: The contribution of the SGT is a bit marginal.**
>
> **A-Weak3:** The proposal of SGT focuses more on improving the robustness of models in the application process. Because the real layout analysis scenario cannot be as clean as the open-source research benchmark. For a Word document, it is unavoidable to make subtle transformations to the position of text blocks, or the size of an image.  As shown in Fig.6 in the Appendix, there are prediction errors due to random slight movement of some text positions. Therefore, we propose using the SGT method to simulate the subtle change of documents to improve the generalization during the training process.
>
> Different from most existing image data augmentation schemes that rely on whole images, our proposed SGT utilizes a hierarchical strategy that divides into three different dimensions: replacing text at sentence-level, scaling and translating at paragraph-level, and cropping at page-level.
>
> In addition, this method does not introduce any parameters yet enhances the data diversity during the training process. It is effective in improving the robustness of the model and is more beneficial for practical application.
>
> **Q1: The fusion strategy only modifies the original image where there is text. What about non-text elements, such as tables or figures? What is the impact of the method on those regions?**
>
> **A-Q1:** In fact, the detection of table and figure elements relies more on visual features rather than semantics. But we find that introducing semantic features not only improves the accuracy of semantically strongly correlated elements, like Text, Title, Section-Header, etc, but also the detection performance of figures and tables. As shown in Fig.1, the accuracy for each category has been improved by our proposed method. This validates the effectiveness of our method.
>
> **Q2: In equation (8), is the token embedding w_m the result of attention on word graph? Or is it just the original token embedding without attention?**
>
> **A-Q2：** It's the 3-dim embedding projected from the original 768-dim token embedding. The detailed desrciption is as previous answer.
>
>
> **Again, thank you very much for the kind efforts in evaluating and helping to improve the quality of our manuscript!**

---

> > ### Comment · Reviewer_ssuv · 2024-11-19
> >
> > Thank you for your response. It helped to clarify some of the questions and the contribution of each of the components of the proposed approach. I have no further questions/comments

---

> > > ### Author Response · Authors · 2024-11-20
> > >
> > > Thank you for providing the constructive comments.
> > >
> > > We will put the descriptions and experiments mentioned above in the original paper. If you have any other questions, please feel free to contact us.
> > >
> > > Best regards,
> > >
> > > Authors of Submission8525

---

> > > ### Author Response · Authors · 2024-11-22
> > >
> > > The modified contents have been presented in the full paper. If you have any other questions, please feel free to contact us at any time.
> > >
> > > Thank you!

---

> ### Author Response · Authors · 2024-12-02
> **A friendly reminder on re-evaluating the value of this paper**
>
> Dear Reviewer,
>
> We sincerely appreciate the discussion, which is vital to enhance the quality and persuasiveness of the paper.
>
> Based on our previous discussion, the main revisions to the original paper are as follows:
>
> 1. We have conducted an ablation experiment on the textual dimension of GMDF. We find that there is almost no difference in accuracy when projecting 768 dimensions to 3 dimensions and 128 dimensions, but 3 dimensions bring almost no computational burden. This confirms our previous intention: it's to use graphs and attention mechanism to construct the connections between words and sentences, the dim for data fusion does not need to be very large, because the backbone will extract high-level features based on the connections.
>
> 2. We have conducted detailed ablation experiments on different modules in GMDF, including WordEmbedding, SentenceGraph, and WordGraph.
>
> 3. We have provided a more detailed description of token embedding w_m.
>
> **From our previous discussions, it seems that our revisions and explanations have resolved your questions.  As the deadline for discussion is approaching, could you take a look at the updated paper, and re-evaluate its value?**
>
> If you have any follow-up concerns, please let us know and we will be happy to answer them.
>
> Best regards,
>
> the Authors

---

### Official Review · Reviewer_tgtr · 2024-11-03

**Soundness:** 3
**Presentation:** 3
**Contribution:** 2
**Rating:** 6
**Confidence:** 4

**Summary:**

This paper presents an architecture for learning a representation of document images. The presented model is addressed to improve the efficiency of Document Layout Analysis systems. It is presented as a pre-processing strategy that can be plugged in baseline methods for semantic segmentation. The paper taxonomizes existing methods for DLA as Vision-based and Multi-modal models. The proposed contributions are: First, the Graphic Multi-modal Data Fusion model uses Graph Attention Networks to generate image features at pixel level that combine word, sentence embeddings with attention scores between them as a structural representation. The second proposal is the  Spatial Geometric Transformation, that consists in several document operations that allow to augment the data used for training, boosting the layout features. Both strategies are integrated in pre-processing steps in the experimental setup consisting in FasterRCNN and CascadeRCNN as baselines.

**Strengths:**

A multimodal system is presented, that combines visual information, semantic embeddings from words and senteces, and an attention mechanism capturing the relative relationships between text elements.

The Spatial Geometric Transformation is a simple but effective method consisting in basic operations that allow to augment the original data with plausible document new instances.

The proposed representation is experimentally shown as an improvement when it is used as a pre-processing step in classical baselines.

**Weaknesses:**

The use of structural information, in particular graph-based representations is not novel. There are several works that combine visual, textual and structural features. In particular, the attention mechanism that is proposed is the classical implementation of the Graph Attention Networks (GAT).

In a layout, there are other components than just text. The strategy of modeling relationship between elements is baed on text words and sentences, but not considering other elements like figures, images, tables... The method seems to be highly sensitive to the detection of text words and lines, disregarding a more macroscopic representation.

**Questions:**

How does your method differ from a classical Tranformer model, that actually captures the attention between words of a text? The attention mechanism implemented between sentences resembles a Graph Transformer architecture.

It is not clear to me how sentences are obtained. Are they "meaningful" sentences obtained by the BERT model? or just word lines?

The extraction of edges in eq (3) is based in the 4 nearest sentences, in terms of their position. Have you considered other strategies, like a visibility graph?

======================== AFTER THE REBUTTAL

After interacting with the authors, and looking at the other reviewers' revisions, I will keep the score of the first review. I thank the authors for their clarifying responses, and the effort to consider the comments. The authors have solved most of my concerns. However, I consider that the proposed method and contribution still has room for improvement in a more solid work, and it prevents me to raise the score.

---

> ### Author Response · Authors · 2024-11-18
>
> **We would like to express our sincere thanks for the constructive comments and suggestions from the reviewer.**
>
> **Weak1: The use of structural information, in particular graph-based representations is not novel. There are several works that combine visual, textual and structural features. In particular, the attention mechanism that is proposed is the classical implementation of the Graph Attention Networks (GAT).**
> **Q1: How does your method differ from a classical Tranformer model, that actually captures the attention between words of a text? The attention mechanism implemented between sentences resembles a Graph Transformer architecture.**
>
> A-Weak1&Q1:
>
> As shown in Fig.2, current layout analysis tasks are primarily divided into models based solely on visual modalities and those based on multimodal approaches. Methods relying on pure visual modalities treat the task as an object detection task, overlooking the fact that documents are rich in textual semantic information. In contrast, multimodal methods take into account the textual characteristics of documents. However, the drawback of those multimodal methods is their excessive computational load. Previous multimodal practices, such as VGT and VSR, essentially follow a multimodal feature fusion approach, treating text information as a grid map. This information is processed through a separate backbone to extract deep-level features before fusion, which leads to a multiplicative increase in computational requirements.
>
> Considering these issues and analyzing that the categories are largely determined by relative relationships, we construct a graphic method for multimodal fusion. Moreover, to prevent the computational load from becoming too large, our approach first performs multimodal fusion based on images and establishes associations, then extracts features with backbone. This has two advantages: compared to previous multimodal layout analysis models, our method introduces a minimal increase in computational load yet surpassing their mAP on multiple Benchmarks. Additionally, since our method integrates features during the pre-processing stage, it can be combined with multiple models without altering the original model architecture, making it a pluggable solution for application.
>
> In terms of algorithmic innovation, we build a graph structure that considers the interrelation between global and local information specific to documents, which can be represented as "Graph in Graph." We construct a Sentencegraph by treating each sentence as a node. The construction of sentence node information comes from two aspects: Firstly, each sentence has textual features, which include a multitude of words. We can obtain word embeddings through a tokenizer. Additionally, each sentence contains image features, such as font color, size, bold and length. Therefore, we further construct an internal Wordgraph inside each sentence, treating tokens within the sentence as nodes and combining the corresponding image and textual information. This structure is then expanded to a larger sentence graph. Both Sentencegraph and Wordgraph rely on attention mechanisms for their computations, but Sentencegraph considers interacting with external information, while Wordgraph considers generating internal representations of sentence.
>
> We have also compared the accuracy when not incorporating the internal Wordgraph, which results in a significant decline in mAP (82.1 ----> 81.3), as shown in the table below.
>
>
> | Detector     | Backbone  | WordEmbedding | SentenceGraph | WordGraph | mAP |
> | ------------ | --------- | --------      |--------       |--------   |-----|
> | Cascade-RCNN | DiT       |              |               |           |  78.0   |
> | Cascade-RCNN | DiT       |      Y         |        Y       |           |  81.3   |
> | Cascade-RCNN | DiT       |      Y         |        Y       |     Y      |  82.1   |

---

> ### Author Response · Authors · 2024-11-18
>
> **Weak2: In a layout, there are other components than just text. The strategy of modeling relationship between elements is baed on text words and sentences, but not considering other elements like figures, images, tables... The method seems to be highly sensitive to the detection of text words and lines, disregarding a more macroscopic representation.**
>
> A-Weak2: In fact, the detection of table and figure elements relies more on visual features rather than semantics. But we find that introducing semantic features not only improves the accuracy of semantically strongly correlated elements, like Text, Title, Section-Header, etc, but also the detection performance of figures and tables. As shown in Fig.1, the accuracy for each category has been improved by our proposed method. Therefore, this method is not just aimed at improving objects containing text, but a universal method for enhancing the effectiveness of document layout analysis.
>
>
> **Q2: It is not clear to me how sentences are obtained. Are they "meaningful" sentences obtained by the BERT model? or just word lines?**
>
>
> A-Q2: We first utilize OCR tool to recoginze characters and obtain the position of each sentence and word. Then, we use the Bert tokenizer to obtain the embedding corresponding to each word. After that, we merge the corresponding images and embeddings as word nodes' information, and aggregate the information of these words through Wordgraph. Finally, we average the words information as the 6-dimensional feature of each sentence. This is what we describe as obtaining the "meaning" of each sentence with local fine-grained representation. We set each sentence as a node for sentencegraph and establish the connection.
>
>
>
>
> **Q3: The extraction of edges in eq (3) is based in the 4 nearest sentences, in terms of their position. Have you considered other strategies, like a visibility graph?**
>
> A-Q3: The main contribution of this paper is to demonstrate that establishing connections between texts during the preprocessing stage is important and using graphs for multimodal fusion is an effective and efficient way to improve layout analysis performance. The experimental results have shown the effectiveness of this method. Other Advanced Graph Theory methods for building graphs will also be a focus of our future research to prompt the research on layout analysis tasks. Besides, not only for layout analysis tasks, it is also interesting to verify that extend the idea of using graph for multimodal information fusion to more document tasks and scenarios with large document models. This is also our future research focus.
>
>
> **Again, thank you very much for the kind efforts in evaluating and helping to improve the quality of our manuscript!**

---

> ### Author Response · Authors · 2024-11-22
>
> The modified contents have been presented in the full paper. If you have any other questions, please feel free to contact us at any time.
>
> Thank you!

---

> > ### Comment · Reviewer_tgtr · 2024-11-24
> >
> > Dear authors,
> > thank you for your exhaustive answer to my comments. I have carefully read the rebuttal report addressed to all the reviewers. It helps to clarify some points that were not clear in the original submission. At this point, I do not have further questions and comments.

---

> > > ### Author Response · Authors · 2024-11-25
> > >
> > > Thank you again for your valuable feedback that has helped us improve the quality of this paper!

---

### Meta-Review · Area_Chair_PzEc · 2024-12-18

**Metareview:**

This paper describes pre-processing strategies to enhance document layout analysis as a crucial step in automated document understanding. The authors point out critical issues with existing image-centric approaches and propose a Graphical Multi-modal Data Fusion technique to connect textual segments by leveraging their relative relationships. Additionally, a Spatial Geometric Transformation strategy is introduced to improve  robustness across varying document structure. These methods aim to improve accuracy while adding minimal computational overhead and are compatible with existing models. Experimental results are given on multiple datasets.

Reviewers variously articulated the following strengths of the proposed approach:
+ **Simplicity and Effectiveness of the Spatial Geometric Transformer**: Spatial Geometric Transformation is a simple, but effective, way of combining basic operations to augment the original data with new, plausible document instances.
+ **Plug-and-Playability**: The representation proposed by the authors can be used as a pre-processing step in classical baselines, leading to improved results with used in combination with Faster-RCNN and Cascade-RCNN models.
+ **Efficiency**: The combination of Spatial Geometric Transformation and Graphical Multi-modal Data Fusion performs as well as the state-of-the-art, but at about half the computational costs in FLOPs.

However, reviewers also articulated a number of weak points that outweigh the strengths of the work:
+ **Insufficient analysis and Motivation for Performance Improvements**: While the performance and efficiency improvement is empirically quantified, there is little analysis or motivation given in terms of *why* the proposed approach yields the empirically observed results. Ablations were added in rebuttal, but a much deeper analysis is needed to justify the proposed approach -- especially in light of the relatively marginal novelty of the proposed system (see below).
+ **Reliance on Word Graphs**: The use of word graphs derived from OCR is questionable, as reviewers point out that there is much more to document layout than spatial arrangement of text. Sensitivity to missed text detection due to OCR failures is not addressed.
+ **Novelty**: Reviewers noted that the Spatial Geometric Transformation proposed is a standard Graph Attention Network, and that the combination of combining visual, textual, and structural information for layout analysis is very standard. As such, the proposed approach is a combination of well-known techniques and represents little novel contribution.

The consensus after the rebuttal period is that the work is in need of significant revision and reformulation before it can be considered for acceptance.

**Additional Comments On Reviewer Discussion:**

There was significant back-and-forth between three of the reviewers and the authors during the discussion phase. The authors provided clarifications and some new experimental results, but the consensus towards Rejection remained after the discussion. The main issues regard novelty and insufficiently deep analysis of the motivations behind the contribution (i.e. no analysis beyond improved empirical results).

---

### Decision · Program_Chairs · 2025-01-22

Reject